# Structures of Native Doublet Microtubules from *Trichomonas vaginalis* Reveal Parasite-Specific Proteins

Alexander Stevens[1,2,3,4], Saarang Kashyap[1,2,4], Ethan H. Crofut[1,2], Shuqi E. Wang[1], Katherine A. Muratore[1], Patricia J. Johnson [1] ✉ & Z. Hong Zhou [1,2,3] ✉

Doublet microtubules (DMTs) are flagellar components required for the protist *Trichomonas vaginalis* (*Tv*) to swim through the human genitourinary tract to cause trichomoniasis, the most common non-viral sexually transmitted disease. Lack of structures of *Tv*'s DMT (*Tv*-DMT) has prevented structure-guided drug design to manage *Tv* infection. Here, we determine the 16 nm, 32 nm, 48 nm and 96 nm-repeat structures of native *Tv*-DMT at resolution ranging from 3.4 to 4.4 Å by cryogenic electron microscopy (cryoEM) and built an atomic model for the entire *Tv*-DMT. These structures show that *Tv*-DMT is composed of 30 different proteins, including the α- and β-tubulin, 19 microtubule inner proteins (MIPs) and 9 microtubule outer proteins. While the A-tubule of *Tv*-DMT is simplistic compared to DMTs of other organisms, the B-tubule of *Tv*-DMT features parasite-specific proteins, such as *Tv*FAP40 and *Tv*FAP35. Notably, *Tv*FAP40 and *Tv*FAP35 form filaments near the inner and outer junctions, respectively, and interface with stabilizing MIPs. This atomic model of the *Tv*-DMT highlights diversity of eukaryotic motility machineries and provides a structural framework to inform rational design of therapeutics against trichomoniasis.

*Trichomonas vaginalis* (*Tv*) is a flagellated, extracellular parasite of the human genitourinary tract and causative agent of trichomoniasis, the most common non-viral sexually transmitted infection (STI), with 250 million infections per annum and global prevalence over 3%[1–3]. *Tv* infection is linked to increased rates of preterm delivery and mortality, genitourinary cancers, and HIV transmission, with a disproportionate impact on women in developing countries[1–5]. Though the antibiotic metronidazole can be curative, concerns about its carcinogenic potential, increasing metronidazole resistance in *Tv*, and rising frequency of reinfection underscore the need for alternative precision therapies[1,6–8]. *Tv* relies on its four anterior and one membrane-bound, recurrent flagellum to propel itself through the genitourinary tract and attach to the mucosa of its human hosts, making the mechanisms driving locomotion potential therapeutic targets[9]. Unfortunately, no

high-resolution structures related to *Tv* flagella are currently available, and even tubulin remains uncharacterized in *Tv* despite a putative role in antimicrobial resistance[10–12].

As observed in low-resolution, thin-section transmission electron microscopy (TEM) studies[13], the locomotive flagella originate from cytosolic basal bodies and extend into the flagellar membrane with decorations along the microtubule filaments that stabilize the tubules and facilitate intraflagellar transport[14–16]. The flagellar core, or axoneme, conforms to the canonical 9 + 2 arrangement wherein a central pair of singlet microtubules (MTs) is connected via radial spokes (RSs) to nine surrounding doublet microtubules (DMTs), which transduce force generated by dynein motor complexes through the flagella (Fig. 1)[13,17,18]. Studies in other organisms revealed DMTs are coated with different combinations of microtubule inner and outer proteins

[1]Department of Microbiology, Immunology & Molecular Genetics, University of California, Los Angeles (UCLA), Los Angeles, CA, USA. [2]California NanoSystems Institute, UCLA, Los Angeles, CA, USA. [3]Department of Chemistry and Biochemistry, UCLA, Los Angeles, CA, USA. [4]These authors contributed equally: Alexander Stevens, Saarang Kashyap. ✉e-mail: johnsonp@UCLA.edu; Hong.Zhou@UCLA.edu

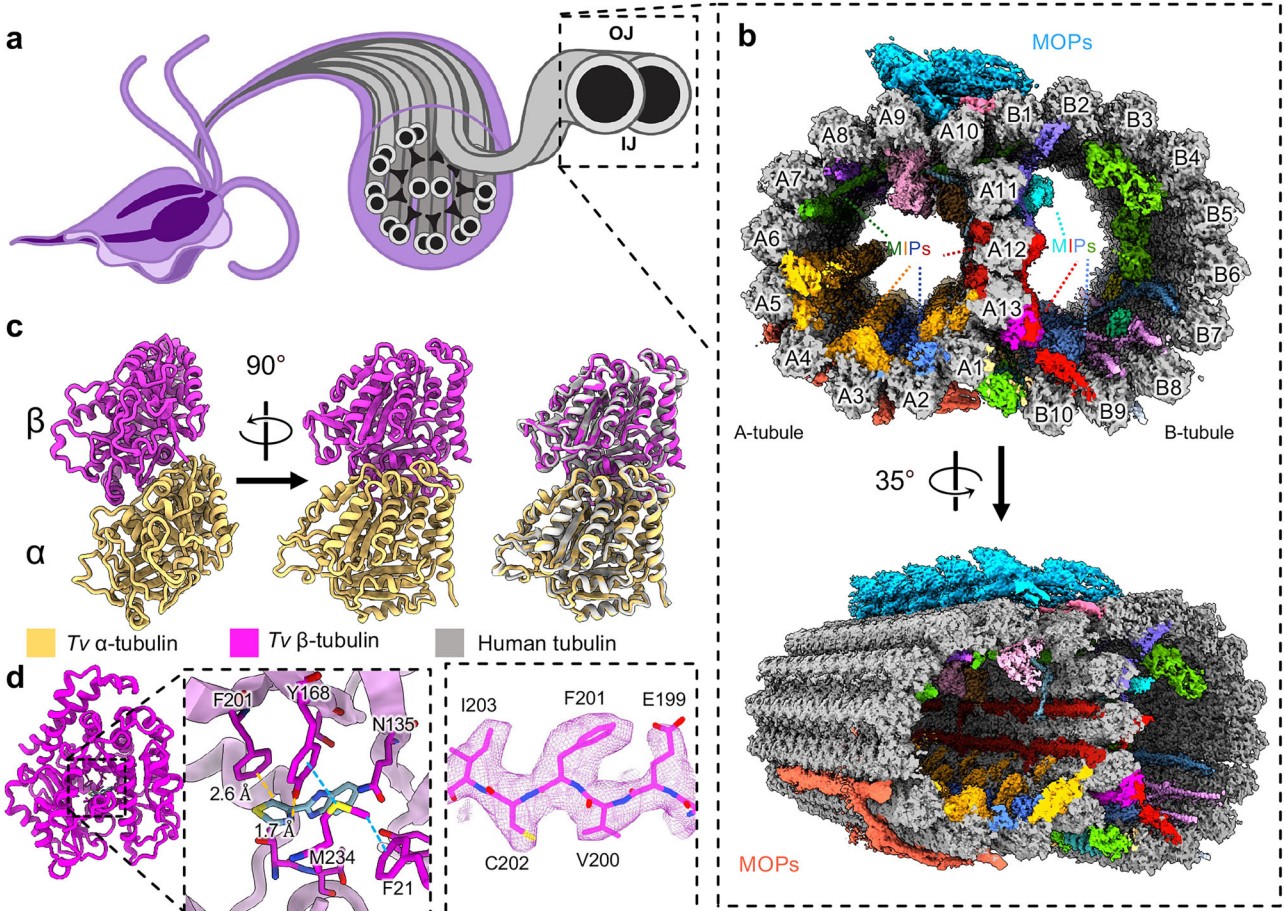

**Fig. 1 | CryoEM reconstruction of the doublet microtubules from *Tv*. a** Diagram of axoneme from the flagella of *T. vaginalis*. **b** Cross-section of *Tv*-DMTs with microtubule inner proteins (MIPs) and microtubule outer proteins (MOPs) indicated with various colors. A- and B-tubules, as well as protofilaments, are labeled. **c** Atomic models of α- (gold) and β-tubulin (magenta), superimposed with human tubulin (gray) to the right. **d** Alternate view of *Tv* β-tubulin (left) and docked thiabendazole molecule (blue) fit into the putative binding site with adjacent residues shown (center) and cryoEM map density (right). **IJ:** inner junction; **OJ:** outer junction.

(MIPs and MOPs) that facilitate assembly, stability, and function (Fig. 2a)[15,19–24].

Dozens of MIPs and MOPs have been identified across numerous studies of eukaryotic flagella, of which about half are conserved[15,19–22]. DMTs from multicellular eukaryotes incorporate more complex MIP arrangements, particularly along the highly variable ribbon protofilaments (PFs) that compose the inner and outer junctions (IJ and OJ) where the A- and B-tubules meet (Fig. 1a)[15,19–22]. In sperm flagella, filamentous tektin bundles near the ribbon PFs are thought to reinforce the long flagella as they swim through the viscous milieu of the genitourinary tract[25,26]. Though the *Tv* genome lacks tektin genes, the parasite swims through the same environment as sperm, coordinating its much shorter flagella into a distinct beating pattern[27]. Considering these apparent differences, it is unclear how the parasite propagates motion under these conditions, and suggests a species-specific adaptation that may be exploited for therapeutic development.

Here, we leveraged mass spectrometry, cryogenic electron microscopy (cryoEM), and artificial intelligence to analyze the DMTs derived from *Tv* parasites and elucidate the structures of the proteins that compose them. Our structure contains 30 distinct proteins, including the α- and β-tubulin, 19 MIPs, and 9 MOPs. Among these, we identified four *Tv*-specific proteins, including one bound to a ligand not observed in the DMTs of other organisms. This structure of the *Tv* flagellum highlights remarkable simplicity in the species' DMT architecture compared to more complex organisms such as mammals, as

well as other protists like *Tetrahymena thermophila*. Despite this simplicity, *Tv* uses its 5 flagella to traverse the same viscous environment as the more complex, monoflagellated mammalian sperm, suggesting a key to parasite locomotion lies in the short list of *Tv*-DMT proteins.

## Results

### *T. vaginalis* DMTs feature both familiar and species-specific MIPs

We optimized a protocol to isolate DMTs from *T. vaginalis* and limit perturbations to the internal structures, and subjected them to single-particle analysis using cryoEM[28]. The resultant cryoEM maps of the 48 nm repeat DMT had a global resolution of 4.2 Å, and focused refinement improved local resolution to between 3.4 Å and 4.2 Å (Fig. 1b, Supplementary Table 1 and Supplementary Figs. 1a, b, 2). Reconstructions of the 16 nm and 96 nm repeat structures were resolved to 3.8 Å and 4.4 Å respectively (Supplementary Fig. 1b and Supplementary Fig. 2). We also collected mass spectrometry data for our cryoEM sample to produce a library of potential *Tv*-DMT proteins and utilized cryoID[29] to identify most likely candidates for certain map densities. AlphaFold predicted structures served as initial models for atomic modeling of both conserved and species-specific cryoEM map densities[30,31]. From our structures, we identified 30 different proteins, including 19 MIPs, 9 MOPs, and the α/β tubulin of *Tv* (Supplementary Movie 1, Supplementary Table 2 and Supplementary Fig. 3). Of these proteins, 15 MIPs and all 9 MOPs are conserved between *Tv* and

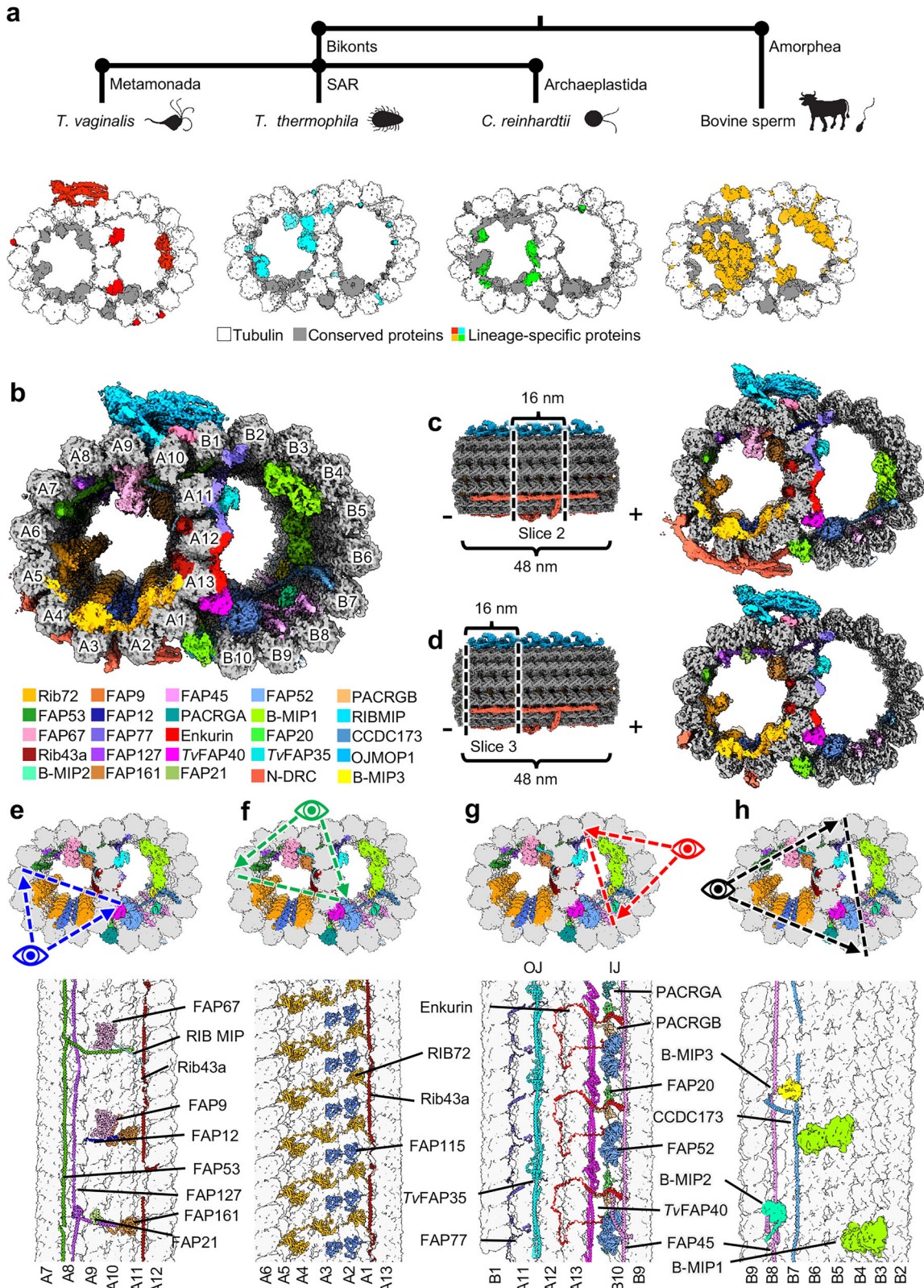

**Fig. 2 | *Tv*-DMTs reveal conserved and species-specific MIPs. a** Phylogeny tree illustrating divergence between Bikonts and Amorphea (top), with example organisms from these branches and accompanying DMTs (bottom) with tubulin (white), conserved flagella associated proteins (FAPs) (gray), and species-specific FAPs (colored red for *Tv*, green for *Clamydomonas*, cyan for *Tetrahymena*, and orange for *Bovine*). **b** Cross-sectional view of cryoEM reconstruction of 48 nm repeat with MIP protein densities colored to demonstrate arrangement. **c**, **d** Cross-sectional view of DMTs from the 48 nm repeat map, shown as different 16 nm long sections throughout the DMT. **e**–**h** Cross-sectional views of *Tv*-DMTs from different perspectives to illustrate MIP arrangement and periodicity.

previous DMT structures, whereas 4 MIPs are *Tv*-specific. There are also 4 unassigned MIP and 3 MOP densities that appear to play an important role in DMT function, but for which we lacked sufficient resolution to model atomically.

Consistent with their ~80% sequence identities, the atomic models of *Tv*'s α- and β-tubulin are nearly identical to those of their human homologs (Fig. 1c), including the region of β-tubulin where many antiparasitic, benzimidazole-derived drugs (BZs) bind (Fig. 1d). Previous studies in *Tv* suggest mutations to aromatic residues Tyr168 and Phe201 in β-tubulin confer BZ resistance[12,32,33]. Indeed, like human β-tubulin's Phe169 and Tyr202, *Tv* orients Tyr168 and Phe201 into the BZ binding pocket where they are stabilized by Aro-Met-Aro interactions with adjacent Met234 and Phe21 residues and sterically occlude BZ drugs like thiabendazole (TBZ) (Fig. 1d). To corroborate this, we performed docking experiments using AutoDock Vina and found TBZ docked β-tubulin produced large positive binding free energy values (ΔG) (Supplementary Fig. 4a, b). By contrast, a virtual β-tubulin Y168F mutant, reflecting the putative TBZ binding site from susceptible organisms like *Ascaris lumbricoides*, exhibited a significantly lower binding free energy when TBZ was docked (Supplementary Fig. 4c). Interestingly, we observe the swapped positions of phenylalanine and tyrosine residues between human and *Tv* β-tubulin, which may help to explain species-specific sensitivity to different BZs.

Like those in other organisms, the α/β tubulin heterodimers polymerize and assemble into rings of 13 and 10 PFs that compose the A- and B-tubules, respectively (Fig. 2b). Within the A-tubule, molecular rulers FAP53, FAP127, and Rib43a impose a 48 nm MIP periodicity and facilitate the organization of other MIPs like FAP67 and RIB72 (Fig. 2e–h). Consistent with studies in *T. thermophila*[15], FAP115 repeats every 32 nm and creates a mismatch with the 48 nm periodicity of the ruler proteins, leading to 96 nm periodicity (Fig. 2f). Interestingly, FAP141 in other organisms is replaced by the smaller *Tv*FAP12, which lashes FAP67 to the A-tubule lumen like the N-terminal helices of FAP53 and FAP127 (Fig. 2e)[19]. Along with the N-terminal segments of FAP53 and FAP127, *Tv*FAP12 spans across the B-tubule to maintain a 16 nm repeating crosslink between the A- and B-tubules[19]. Unlike other species, the *Tv* ribbon PFs (A11-A13) that divide A- and B-tubules are sparsely decorated with A-tubule MIPs, suggesting alternative strategies of ribbon arc stabilization.

In the B-tubule lumen, we found assembly-related MIPs FAP45, CCDC173, Enkurin, FAP77, FAP52, FAP20, and PACRGA/B that are conserved amongst other organisms. Along the B-tubule side of the ribbon arc, we identified the filamentous MIPs *Tv*FAP35 and *Tv*FAP40, which run lengthwise along the A11 and A13 PFs respectively and may compensate for the dearth of MIPs along the ribbon arc in the A-tubule lumen (Fig. 2g). Further, we observed globular MIPs that span PFs B3-B4 and B5-B6 and exhibit 96 nm periodicity (B-MIP1) (Fig. 2h). While the map resolution was insufficient to model these proteins, their interactions with neighboring ruler proteins like CCDC173 indicate an enforced periodicity of 96 nm, which is the largest repeating periodicity of any DMT MIP to date. Though we observed several species-specific proteins, the *Tv*-DMTs have the simplest MIP organization in the A-tubule with just 10 MIPs (nine identified and one unidentified) compared to the next simplest species of record, *Chlamydomonas reinhardtii*, with 22 A-tubule MIPs[19]. This comparatively simple MIP organization in *Tv* suggests the few *Tv*-specific MIPs may play a substantial role in flagellar function.

### *T. vaginalis* microtubules decorate the inner junction with species-specific protein

The DMT IJs of other organisms are typically composed of FAP52, Enkurin/FAP106, PACRG isoforms (PACRGA and PACRGB), and FAP20, while *T. thermophila* and mammalian DMTs include globular proteins atop FAP52 that mediate interactions with PF A13[15,26,34]. Interestingly, the *Tv*-DMT cryoEM map revealed the long, filamentous protein

*Tv*FAP40, running atop PF A13 at the IJ, which alters the topography of this important protofilament. *Tv*FAP40 monomers repeat every 16 nm and are arranged head-to-tail, where head-tail polarity corresponds to the − and + ends of the DMT, respectively (Fig. 3B, C). Each *Tv*FAP40 monomer consists of a globular N-terminal 'head'-domain (residues 1–145) connected to a coiled-coil 'tail' (residues 149–361). The tail consists of 3 helices (α7-9) where a proline-rich kink connects α7 and α8, while a 180° turn at the linker between α8 and α9 forms the 'tip' of the tail. The tip includes α8 and neighboring residues of α9 (residues 246–282), with both a polar face oriented towards the MT and a hydrophobic face oriented towards a neighboring *Tv*FAP40 monomer (Fig. 3c, d). As the kink reaches into the cleft between tubulin heterodimers, α8 is brought into close contact with tubulin and establishes electrostatic interactions. The kink also offsets α7 from α8, creating an overhang to bind the head of a neighboring *Tv*FAP40 monomer, which may help stabilize the interaction (Fig. 3d).

*Tv*FAP40's location along PF A13, unlike any other known MIP, alters conserved MIP interactions at the inner junction. In other organisms, the PACRGB N-terminus binds the groove between A11 and A13, but in our structure, *Tv*FAP40 blocks this groove and replaces A13 as the binding partner. Enkurin, which connects PFs B9 and B10 to A12 and A13 and is critical for DMT assembly and stability[26,35], crosses over the coiled-coils of *Tv*FAP40 near its head domain. In addition, the *Tv*FAP40 C-terminus hooks around α2 of Enkurin, where *Tv*FAP40's C-terminal tyrosine (Tyr377) creates a hydrophobic interface with adjacent aromatic residues from Enkurin (Tyr155 and Trp156) (Fig. 3f). This C-terminal hook acts in concert with the *Tv*FAP40 head that binds the other side of Enkurin α2 and restricts it such that the bottom end of the helix is 1 nm closer to A13 than in other structures.

### *Tv*-specific FAP40 head domain binds an unidentified ligand

In addition to binding neighboring monomers, *Tv*FAP40 incorporates a ligand-binding pocket. Our cryoEM maps indicate the *Tv*FAP40 head domain binds a star-shaped ligand, and our atomic model indicates this pocket is positively charged (Fig. 4a–c and Supplementary Movie 2). Indeed, the putative binding site features seven positively charged side chains oriented towards the points of the star (Fig. 4c), which suggests negatively charged functional groups (Fig. 4c–g). Sequence and structural homology searches within UniProt or the RCSB protein database could not identify similar binding sites[36,37]. However, analysis in PUResNet predicts this to be a ligand binding site[38].

The star-shaped density had apparent stereochemistry reminiscent of the functional groups from a non-planar six-membered ring (Fig. 4h). To narrow the ligand candidates, we used the cryoEM ligand classification software LigandRecognizer to determine what class of ligand our density most likely belonged to (Supplementary Table 3)[39]. Among the hits, we found the sugar ring of the β-d-mannose-like (NGA-like) class of sugars to fit the density's shape (Supplementary Fig. 4g) (Supplementary Table 3). However, the functional groups within this class are too small to occupy the density observed in our map (Supplementary Fig. 4g). Unfortunately, many sugars are poorly represented in the PDB, so unambiguous identification is impractical at near-atomic resolution, and the top hit within the ligand classification job suggested a "rare ligand" (Supplementary Table 3). Based on the character of the binding site and shape of our ligand density, we suspected that it might be a carbohydrate with negative electrostatic potential. Inositol pentakisphosphate (IP5) proved to be both a good fit for the ligand density and exhibited a favorable predicted free energy of binding (ΔG) using Autodock Vina (Supplementary Table 4). Using Autodock Vina to screen a library of potential ligands from a recent metabolomics study of *Tv*[40], we found IP5 exhibited among the lowest ΔG values (Supplementary Table 4). Further, the docked IP5 molecules had conformations similar to that of the molecule fit into the density. (Supplementary Fig. 4e, f). These results support the notion that an inositol phosphate-like carbohydrate molecule may act as a ligand within the *Tv*FAP40 head domain.

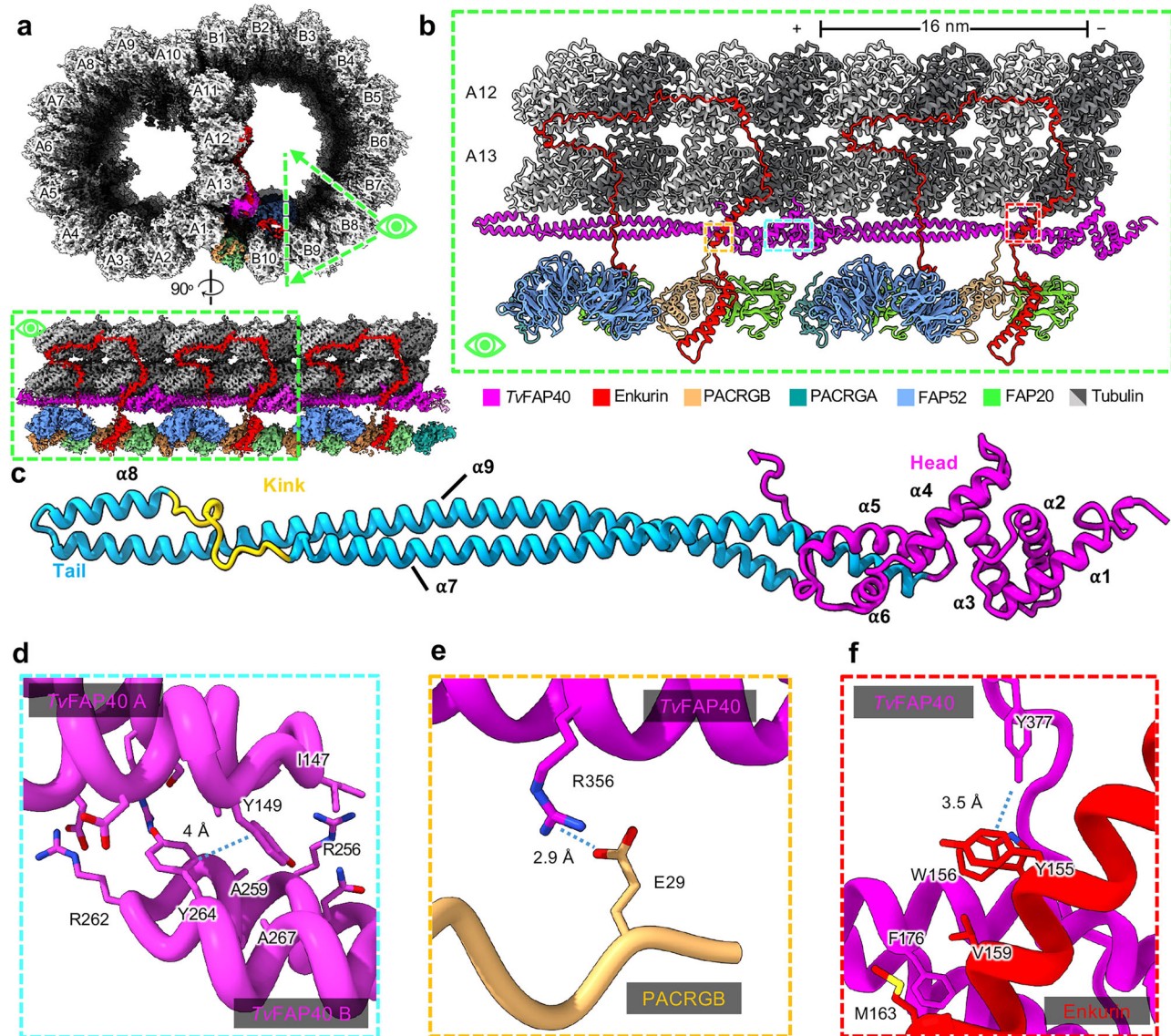

**Fig. 3 | *Tv*FAP40 alters the inner junction arrangement in parasite DMTs.**
**a** Cross-sectional view of cryoEM reconstruction of 16 nm repeat with protofilaments labeled and proteins near the inner junction colored (top) and a cutaway view of a region of interest (bottom). **b** View of atomic models built from the map in

(**a**). **c** atomic model of *Tv*FAP40 colored by domain. **d** Zoomed-in view of the dimerization domain between two *Tv*FAP40 monomers (labeled *Tv*FAP40 A and B). **e**, **f** Close-up view of interaction between PACRGB (tan) and *Tv*FAP40 (magenta) and Enkurin (red), with residues shown to highlight interactions.

## *Tv*FAP35 secures FAP77 to the outer junction of *Tv*-DMTs

In the B-tubule above TvFAP40, we identified a filamentous density along the ribbon PF A11 as *Tv*FAP35, another *Tv*-specific protein (Fig. 5a–c). In the B-tubule, *Tv*FAP35 repeats every 16 nm in a head-to-tail fashion with the heads and tails oriented to the – and +-ends, respectively (Fig. 5b). The *Tv*FAP35 tail domain has the same 'kinked-coiled-coil' fold as *Tv*FAP40, including the tips that mediate MT binding and dimerization (Fig. 5c–f). The head domain of *Tv*FAP35 differs from *Tv*FAP40, as it includes only a flexible N-terminus and helix-turn-helix (Fig. 5c), as opposed to the six helices found in the *Tv*FAP40 head (Fig. 3c).

The position of *Tv*FAP35 along A11 is similar to that of tektin-like protein 1 (TEKTL1), which is thought to reinforce OJ stability during flagellar beating in sperm DMTs[26]. Further, the coiled-coil structure of *Tv*FAP35 resembles the 3-helix bundle architecture of TEKTL1. Unlike TEKTL1, the coiled coils of *Tv*FAP35 include a proline-rich kink that occupies the cleft between tubulin heterodimers. As PFs bend, gaps form at the interface between tubulin heterodimers[41], and the *Tv*FAP35 kink may create stress relief points along A11 by acting as a flexible

linker that accommodates bending. Thus, like TEKTL1, the coiled-coils of *Tv*FAP35 may provide structural stability to the DMT while the kink allows bending and greater flexibility. *Tv*FAP35 also interacts with FAP77, a MIP that aids in B-tubule formation in *T. thermophila* and tethers complete A- and B-tubules together at the OJ (Fig. 5g, h)[15]. In *Tv*, the FAP77 helix-turn-helix motif (residues 140–164) is braced to PF A11 via electrostatic interactions with the coiled-coil of *Tv*FAP35 (Fig. 5b, c, and f). Further, the tail-domain of *Tv*FAP35 passes over residues 238–246 of *Tv*FAP77 which run between a cleft of the A11 PF and reinforces *Tv*FAP77's association to A11 (Fig. 5g). Because FAP77 is implicated in B-tubule assembly[15], the interactions of *Tv*FAP35 with FAP77 and A11 suggest that *Tv*FAP35 may also contribute to DMT assembly.

## *Tv*FAP35 and *Tv*FAP40 proteins share an ancient MT binding motif

Comparison between MIP candidates from other species revealed conserved microtubule binding motifs[42]. Upon comparison, we noticed both *Tv*FAP35 and *Tv*FAP40 have kinked-coiled-coils composed of three helices (α1-3, Supplementary Fig. 5) with similar lengths,

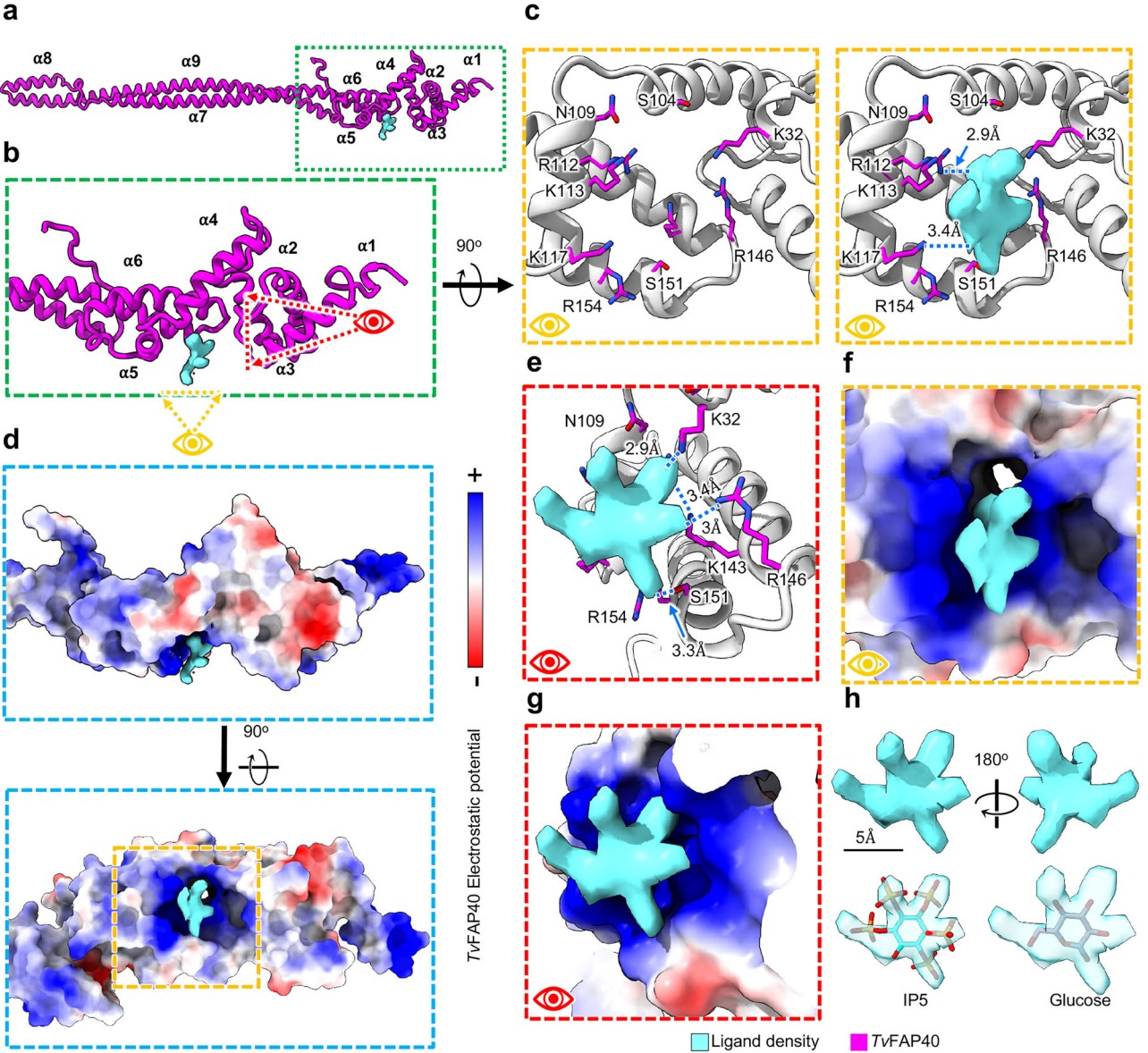

**Fig. 4 | *Tv*FAP40 binds a star-shaped ligand in a positively charged pocket.**
**a** Atomic model of *Tv*FAP40 (magenta) with (**b**) zoomed-in view of the head domain and (**c**) perspectives of the putative binding site with (right) and without (left) ligand density from the cryoEM map. **d** Coulombic potential map of the head domain from B (top) and rotated (bottom) views with blue and red indicating positive and negative Coulombic potentials, respectively. **e** Side-view of ligand in binding pocket with adjacent residues shown. **f**, **g** Views from C and E shown with electrostatic potential maps of *Tv*FAP40 with ligand bound. **h** View of *Tv*FAP40 ligand density (top) with candidate molecules fitted (bottom).

dimerization domains, and MT binding motifs (Figs. 3 and 4). This similarity prompted us to search for homologous proteins via amino acid sequence alignment, but this returned few candidates. Interestingly, the coiled-coils of *Tv*FAP35 and *Tv*FAP40 share just 23% identity, despite similar folds. We next turned to structural alignment using FoldSeek[43] and identified numerous structural homologs (Supplementary Table 5). After curating homolog candidates by removing those without kinked-coiled-coil domains or with TM-scores below 0.4, 31 homolog candidates were selected for further comparison. Initial analysis revealed all kinked-coiled-coil-containing homologs belonged to the group Bikonta, which includes many protists, and excludes animals, fungi, and amoebozoans. Other Bikonta members with these kinked-coiled-coil proteins, like *T. brucei*, similarly lack tektins to stabilize their microtubules against bending forces[44].

Six kinked-coiled-coil homologs (UniProt IDs: A4IAR0, Q38A11, A0A6V1QLJ6, A0E9V2, A0A7J7PGH9, A0A7S3WEM8) from protists representing different clades were selected for multiple sequence alignment with *Tv*FAP35 and *Tv*FAP40, which revealed several conserved residues from the dimerization and MT binding domains (Supplementary Fig. 5a–c). Based on the *Tv*FAP35 sequence, the dimerization domains include hydrophobic residues at Val213 and aromatic residues at Tyr103 and Tyr218, which form hydrophobic interfaces between neighboring monomers (Supplementary Fig. 5g). Further, the MT binding motif on α2 has a high proportion of charged residues, which are likely important in tubulin binding (Supplementary Fig. 5f). Outside of the dimerization and MT binding motifs, the kinked-coiled-coils exhibit an average sequence conservation of ~ 20%, which may be necessary to accommodate different MIPs, as observed in our identified proteins (Figs. 3 and 5).

## *T. vaginalis* microtubule outer proteins exhibit 8 nm periodicities
Considering the marked simplicity of the *Tv*-DMT MIP arrangement, we expected to find a comparably simple MOP organization. Along the

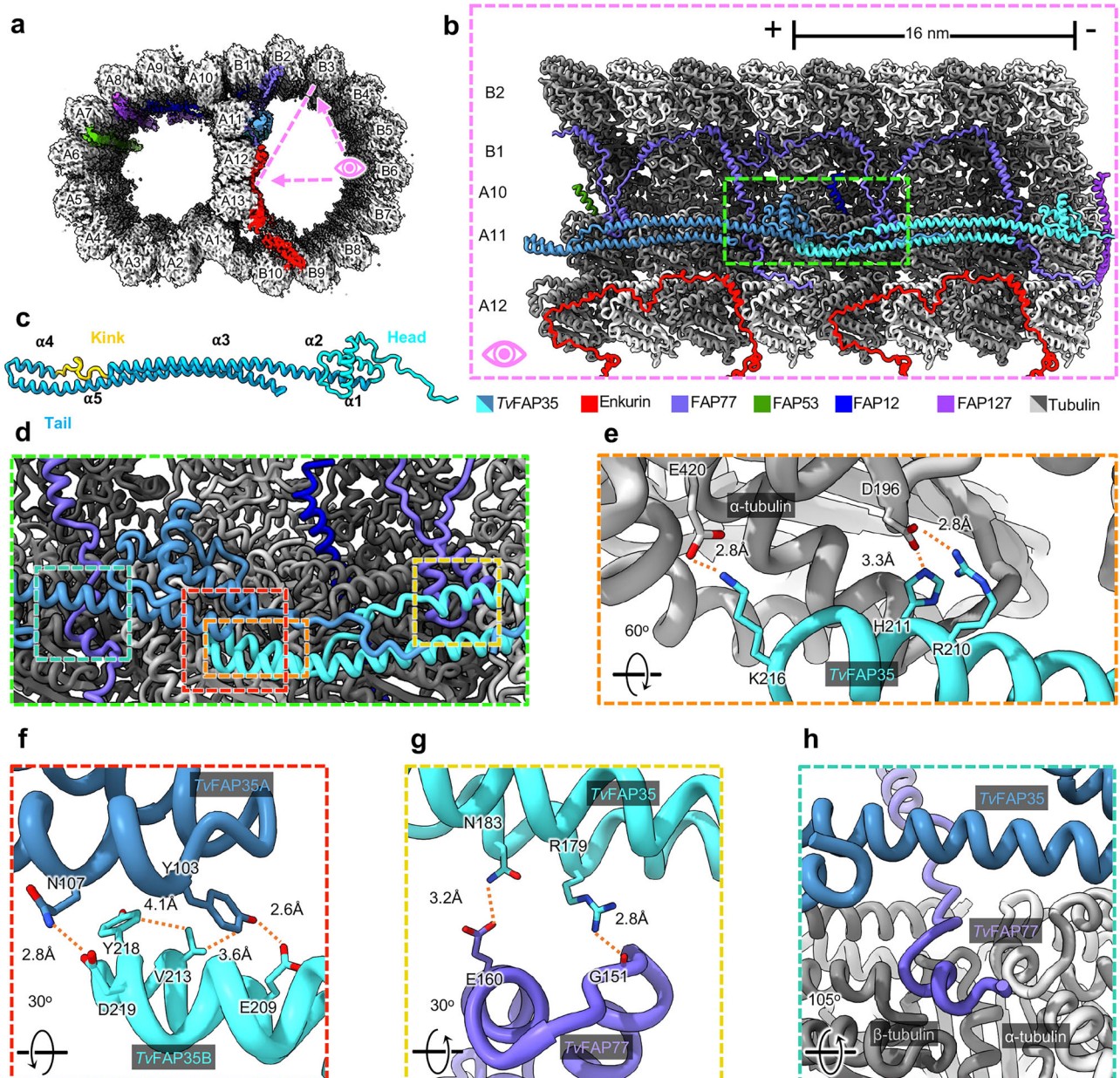

**Fig. 5 | *Tv*FAP35 lines ribbon PF A11 and outer junction proteins. a** Cross-sectional view of the *Tv*-DMT cryoEM map with enkurin and outer junction proteins colored. **b** 32 nm section of protofilaments A10, A11, A12, B1, and B2, along with their associated MIPs, shown with atomic models. **c** *Tv*FAP35 monomer labeled with head (cyan), tail (blue), and kink (yellow), with helix numbers. **d** Zoomed-in view including important interactions of *Tv*FAP35. **e** Electrostatic interactions at the MT-binding motif of *Tv*FAP35. **f** Mixed residue interactions at the dimerization interface between *Tv*FAP35 monomers. **g** Interactions between *Tv*FAP35 and the helix-turn-helix (residues 140–164) of *Tv*FAP77. **h** Residues 238–246 of *Tv*FAP77 pass near the *Tv*FAP35 coiled-coil. Residues 255 and after of *Tv*FAP77, which stretch further down, are omitted for clarity.

A-tubule, we observed both the canonical N-DRC and radial spoke complexes that mediate inter-axoneme connections and flagellar bending (Fig. 6a–c and Supplementary Fig. 6). Like other DMT structures, the axoneme-related proteins exhibit 96 nm periodicity enforced by the molecular ruler proteins CCDC39 and CCDC40, which coil their way between PFs A3 and A2 (Fig. 6b–e). Attached to the molecular ruler proteins, low resolution densities corresponding to the stalk of the radial spoke, inner dynein, and the baseplate of the N-DRC emerge to mediate inter-axoneme connections and facilitate flagellar bending (Supplementary Fig. 6). We identified densities corresponding to FAP251 and FAP91, which contribute to the assembly of the RS3 base, as well as dynein light chain components at the base of the RS2 complex (Supplementary Fig. 6). Furthermore, we identified

homologous dynein regulatory complex (N-DRC) proteins that extend from the bifurcated coiled-coils of the N-DRC to form the stalks of the inner dynein arms (Supplementary Fig. 6). A focused analysis of the radial spoke head-neck complex revealed 11 conserved RS1/RS2-associated proteins including the RSP16 homodimer that connects the two RS heads (Supplementary Fig. 6c). We also identified six densities lacking obvious homologs in previously resolved structures, including distinct globular domains near RSP1 and RSP4 (Supplementary Fig. 6c).

Besides N-DRC and radial spoke proteins, a diverse arrangement of filamentous MOPs occupies the clefts and the surface of several PFs. Previous DMT structures from other species found the shortest MOP periodicity to be 24 nm[15,26]. MOP1 is 24 nm repeating MOP that arranges head-to-tail in the furrows between PFs A3, A4, B8, B9, & B10 and

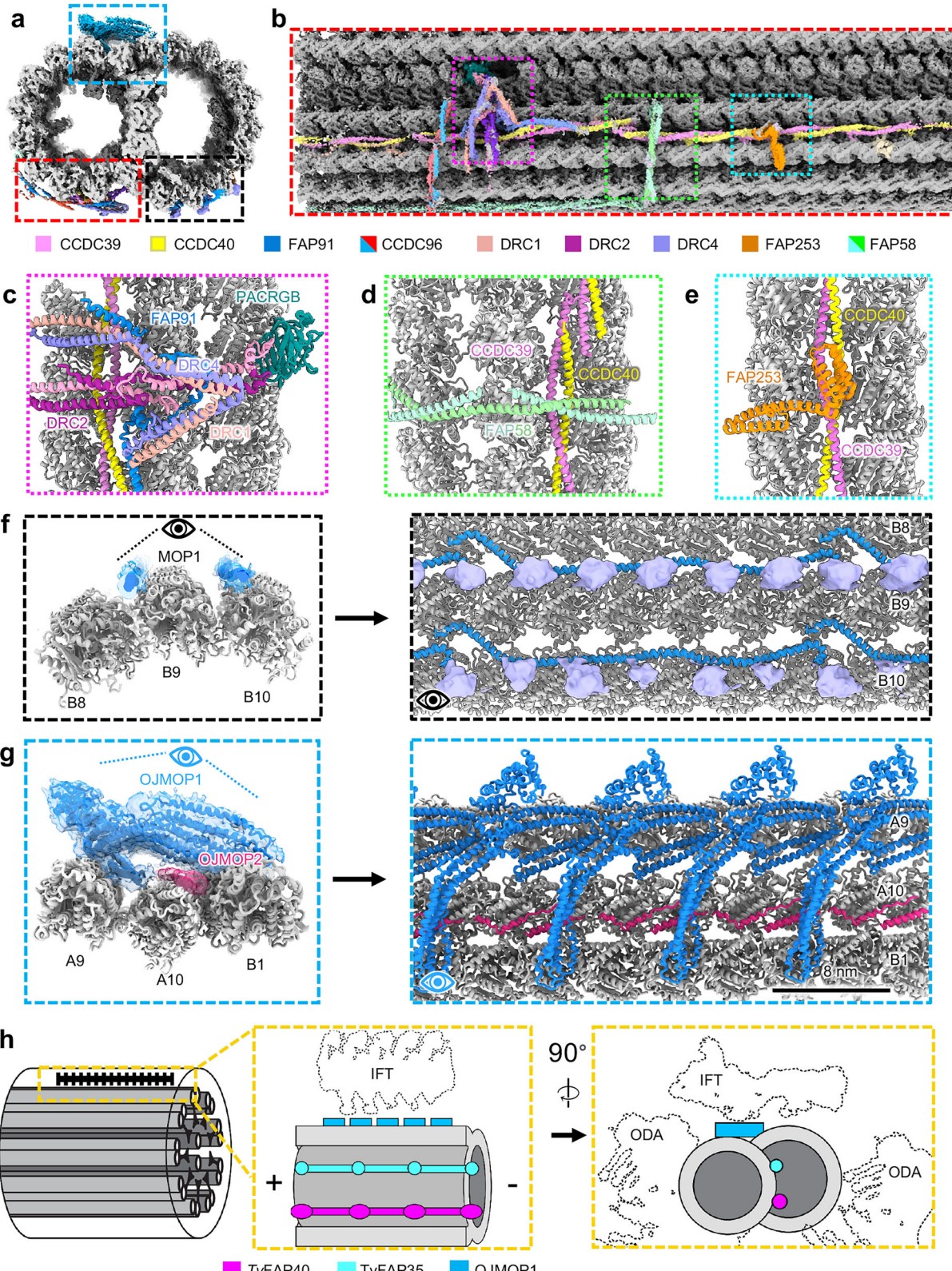

**Fig. 6 | Microtubule organization reveals distinct 8 nm periodicity. a** Cross-sectional view of 96 nm repeat map, colored by different MOP proteins. **b** External view of *Tv*-DMT and zoomed in views of MOPs (**c**–**e**). **f** MOP1 demonstrating 24 nm periodicity as cross-section (left) and external view (right). **g** OJMOP1 (colored blue) and OJMOP2 (colored pink) demonstrating 8 nm periodicity with cross-sectional (left) and external views (right). **h** Schematic view of *Tv*-DMT organization with dotted lines to indicate positions of IFT and inner and outer dynein arm attachment (IDA and ODA).

contacts the flexible C-terminal tails of α and β tubulin in the B-tubule (Fig. 6f). Interestingly, though the exterior of the outer junction is sparsely decorated in DMTs structures from other organisms[19,26]. We found this area to contain a large filamentous protein that repeats every 8 nm and a smaller filament that runs in a zig-zag beneath it and between A10 and B1 (Fig. 6g). The large protein density fashions an ankyrin-like domain seated atop a large coiled-coil domain, which spans the gap between PF A9 and B1 (Fig. 6g).

Due to their limited local resolution, we were unable to confidently assign the identities of these proteins and instead dubbed them *Tv* outer junction microtubule outer protein 1 and 2 (OJMOP1 and OJMOP2) for the large and zig-zag MOPs, respectively. OJMOP1 exhibits an 8 nm periodicity like that of tubulin heterodimers, an unusual repeat length amongst DMT MOPs, and crosslinks PF B1 to A9 and A10 (Fig. 6g). OJMOP1 was observed in only 1/5 of particles, suggesting that some may have been lost during DMT isolation, that OJMOP1 localizes to specific regions of the axoneme, or that different flagella exhibit varying MOP distributions. Exhaustive search through AlphaFold predicted structures from our proteomic data using a strategy similar to that of DomainFit[45] yielded the following 5 candidate proteins, which contain both ankyrin and coiled-coil domains: TVAGG3_0305310, TVAGG3_0421180, TVAGG3_0431750, TVAGG3_0596110, and TVAGG3_0415080. However, none of these candidates could fully account for the observed density, and so it remains unclear if OJMOP1 is composed of one or more of these proteins. Recent work in *C. reinhardtii* has demonstrated that anterograde intraflagellar transport (IFT) brings IFT-B complexes directly over the area where OJMOP1 localizes (Fig. 6h)[46]. However, as IFT components are often lost during DMT isolation, they are unlikely candidates. *As* OJMOP1 features an ankyrin domain oriented towards the would-be IFT-B cargo (Fig. 6g–h), it may interact with TPR-rich proteins of IFT-B to stabilize the cargo. Additionally, others have documented the tendency for cytoplasmic dynein motors to jump between PFs[47]. OJMOP1 may help segregate transport tracks and keep the dynein motors on their preferred A-tubule PFs.

## Discussion

*T. vaginalis* pathogenesis relies on the parasites' locomotive flagella to establish infection and spread between human hosts[27,48]. This study reports the high-resolution structure of the *Tv* flagellar doublet microtubule and its associated complexes, elucidating its molecular composition, architectural arrangement, atomic structures, and small-molecule ligand binding sites. In addition to the structures of the *Tv* tubulin subunits comprising the DMTs, we observed 19 MIPs and 9 MOPs distributed across the A- and B-tubules. Besides several *Tv*-specific proteins, most of these MIPs and MOPs are conserved across ciliated organisms and mediate DMT function in the flagella of these organisms. As the near-atomic structure of flagellar microtubules in the major human parasite *Trichomonas vaginalis*, our results provide a structural framework to understand the parasite's distinct locomotion, offer insights into antibiotic drug resistance, and identify targets for precision medicine.

With a relatively short list of both conserved and *Tv*-specific MIPs, the *Tv*-DMT is perhaps the simplest among known DMT structures. Notably, the *Tv* A-tubule fashions the fewest MIPs of any characterized organism (Fig. 2). Among them, the ruler proteins Rib43a, FAP53, and FAP127 are conserved, but lack many of the interacting partners of their homologs in other species, such as mammalian sperm, that traverse the same environment. The sparsity of A-tubule ribbon proteins in *Tv* suggests these proteins are less essential for locomotion in the human genitourinary tract, which contrasts with the complex MIP arrangement of sperm-specific proteins and tektin bundles seen in the A-tubule of mammalian sperm[26,34]. While *Tv* and sperm exhibit distinct flagellar beating patterns, the sinusoidal beating pattern of the recurrent flagella in *Tv* suggests the additional MIP complexity observed in sperm is not essential to this style of beating. However, human sperm swim five times faster than *Tv* and must propagate beating over a single, much longer flagellum; therefore, the sperm's complex A-tubule MIP arrangement may facilitate sperm's rapid propulsion through the viscous environment, whereas the multi-flagellated *Tv* adopts its own species-specific motion[9,49].

Remarkable specialization is observed in the B-tubule, where several *Tv*-specific proteins line the ribbon arc in a manner similar to tektin bundles from other organisms[19]. Like tektin, the *Tv*FAP35 and *Tv*FAP40 proteins exhibit 16 nm periodicity and similarly interact with other MIPs along their respective protofilaments (Figs. 3 and 5). However, unlike tektin, *Tv*FAP35 and *Tv*FAP40 have variable head domains which seemingly confer different functionalities. To this end, the positively charged pocket of *Tv*FAP40 putatively binds a charged carbohydrate derivative ligand, potentially with an inositol polyphosphate-like shape (Fig. 4). Inositol polyphosphates are abundant cellular polyanions known to stabilize positive interfaces such as the pore of HIV nucleocapsids, and IP5 in particular can act as a pocket factor within the HIV nucleocapsid lattice[50–52]. Considering *Tv*FAP40 interacts with the stabilizing Enkurin protein, ligand binding may ensure proper conformational arrangement of the *Tv*FAP40 head and reinforce the interactions between the monomers at the head-tail interface that present the proper interface for Enkurin. More broadly, our results point to a potentially unrecognized role of carbohydrate derivatives in flagellar function and stability.

The *Tv*-specific MIPs and MOPs are particularly significant in light of their role in propagating the pathogenesis of the most widespread non-viral STI[3]. Specialization differentiates the parasite's DMTs from those of other organisms, including their human hosts, and thus, drugs targeting these specialized components would have minimal toxicity. For instance, the binding pocket found in the *Tv*-specific *Tv*FAP40 protein has a structure with no known homologs and appears to bind a rare ligand (Fig. 4). The distinct shape of this pocket could be leveraged to develop antimicrobial compounds to attenuate parasite DMT function with limited off-target effects. Notably, the only proteins structurally homologous to *Tv*FAP40 and *Tv*FAP35 belong to other Bikonts and include other human-borne parasites like *T. brucei* and *Leishmania infantum*, that may incorporate similar species-specific proteins (Supplementary Fig. 5). While this study represents the distinct DMT from this human-borne parasite, it has demonstrated the power of cryoEM over other structural or in silico methods to open avenues for rational drug design. Together, our findings provide a basis to explore the contribution of microtubule-associated proteins to the distinct aspects that allow *T. vaginalis* to swim through the human genitourinary tract and the diversity of eukaryotic motility in general. Moreover, the atomic details revealed in species-specific proteins and bound small molecules can inform the rational design of therapeutics.

## Methods

### Cell culture

*Trichomonas vaginalis* strain G3 was purchased from ATCC (#PRA-98). *T. vaginalis* strain G3 was cultured in Diamond's modified trypticase-yeast extract-maltose (TYM) medium supplemented with 10% horse serum (Sigma-Aldrich), 10 μ/mL penicillin, 10 μg/ml streptomycin (Gibco), 180 μM ferrous ammonium sulfate, and 28 μM sulfosalicylic acid[53]. 2 L of parasites, grown at 37 °C and passaged daily, were harvested by centrifugation, and washed twice with phosphate-buffered saline, and pelleted at low speed. Cells were resuspended in 50 mL lysis buffer (2% IGEPAL CA-630, 2% Triton X-100, 10% glycerol, 10 mM Tris, 2 mM EDTA, 150 mM KCl, 2 mM MgSO4, 1 mM dithiothreitol [DTT], 1 × Halt protease inhibitors [pH 7.4]) and lysed in a Stansted cell disrupter, operated at 30 lb/in² front pressure and 12 lb/in² back pressure.

Cytoskeletal elements were harvested similar to what has been previously described[28]. Lysates were recovered and maintained at 4 °C for all subsequent steps. Nuclei were removed via low-speed

centrifugation (1000 x g) for 10 min to generate pellet 1 (P1) and lysate 1 (L1). L1 was centrifuged (10,000 × g for 40 min) to pellet cytoskeletal components into P2 and L2. Cytoskeleton pellets (P2) were resuspended in 1 mL low salt (LS) buffer (150 mM NaCl, 50 mM Tris, 2 mM MgCl$_2$, 1 mM DTT, 1× complete protease inhibitor (Sigma-Aldrich) [pH 7.4]) and centrifuged at low speed (1000 × g, 10 min) to pellet cellular debris into P3. The resulting lysate (L3) was placed on a sucrose cushion (30% w/v sucrose in LS buffer) and centrifuged at low speed (1800 × g, 10 min). The supernatant atop the cushion was collected and resuspended in 1 mL LS buffer prior to centrifugation (5000 × g 15 min) to pellet larger cytoskeletal components (P4). The lysate was finally centrifuged at high speed (16,600 × g, 40 min) to pellet axoneme-related cytoskeletal elements (P5). The was then resuspended in a minimal volume of LS buffer supplemented with 5 mM ATP and left at RT for 1 h.

### In-solution digestion, Mass spectrometry data acquisition and analysis

*T. vaginalis* cytoskeleton pellet, P5, resuspended in low salt (LS) buffer, was mixed with 4 × volume of ice-cold acetone and kept at -20 °C for 2 h. The mixture was centrifuged at 4 °C with 14,000 rpm for 15 min and the supernatants discarded. The air-dried pellet was fully dissolved in 8 M Urea in 100 mM Tris-HCl (pH 8) at 56 °C, and the proteins reduced with 10 mM Tris(2-carboxyethyl) Phosphine for 1 h at 56 °C. The reduced proteins were then alkylated with 40 mM iodoacetamide for 30 min in the dark at room temperature, and the reaction was quenched with Dithiothreitol at a final concentration of 10 mM. The alkylated samples were subsequently diluted with 7 × volume of 100 mM Tris-HCl pH 8, to 1 M Urea concentration. To generate peptides, Pierce Trypsin Protease (Thermo Fisher Scientific) was added to the samples, and the ratio of trypsin:protein was 1:20 (w/w). The digestion reaction was incubated at 37 °C overnight, and the residue detergents in the protein sample were removed using a HiPPR Detergent Removal Spin Column Kit (Thermo Fisher Scientific) on the next day. Prior to the mass spectrometry assay, the sample was desalted with Pierce C18 Spin Columns (Thermo Fisher Scientific) and lyophilized.

Three biological replicates were prepared and trypsin-digested following the steps above for P5. The lyophilized protein pellets were dissolved in sample buffer (3% Acetonitrile with 0.1% formic acid) and ~1.0 μg protein from each sample was injected to an UltiMate 3000 RSLCnano (Thermo Fisher Scientific), which was equipped with a 75 μm x 2 cm trap column packed with C18 3 μm bulk resins (Acclaim PepMap 100, Thermo Fisher Scientific) and a 75 μm × 15 cm analytical column with C18 2 μm resins (Acclaim PepMap RSLC, Thermo Fisher Scientific). The nanoLC gradient was 3 − 35% solvent B (A = H2O with 0.1% formic acid; B = acetonitrile with 0.1% formic acid) over 40 min and from 35% to 85% solvent B in 5 min at a flow rate 300 nL/min. The nanoLC was coupled with a Q Exactive Plus orbitrap mass spectrometer (Thermo Fisher Scientific), operated with Data Dependent Acquisition mode (DDA) with an inclusion list for the target peptides. The ESI voltage was set at 1.9 kV, and the capillary temperature was set at 275 °C. Full spectra (m/z 350 - 2000) were acquired in profile mode with a resolution 70,000 at m/z 200 with an automated gain control (AGC) target of 3 × 10$^6$. The most abundant 15 ions were subjected to fragmentation by higher-energy collisional dissociation (HCD) with a normalized collisional energy of 25. MS/MS spectra were acquired in centroid mode with a resolution 17,500 at m/z 200. The AGC target for fragment ions was set at 2 × 10$^4$ with a maximum injection time of 50 ms. Charge states 1, 7, 8, and unassigned were excluded from tandem MS experiments. Dynamic exclusion was set at 45.0 s.

The raw data was searched against total *T. vaginalis* annotated proteins (version G3) downloaded from TrichDB, using ProteomeDiscoverer 2.5 (Thermo Fisher Scientific). The following parameters were

set: precursor mass tolerance ± 10 ppm, fragment mass tolerance ± 0.02 Th for HCD, up to two miscleavages by semi-trypsin, methionine oxidation as variable modification, and cysteine carbamidomethylation as static modification. Protein abundance was quantified using the sum of the three most intense peptides coming from the same protein. The minimum peptide length for identification was set at 6 amino acids, and at least one peptide was required to identify a protein. Proteome Discoverer applied false discovery rate (FDR) thresholds to assess confidence levels: identifications with an FDR ≤ 1% were classified as high confidence, those with an FDR ≤ 5% as medium confidence, and any identifications exceeding this threshold were considered low confidence. At the protein level, the FDR was also controlled at 1% to ensure high-confidence identifications. Only proteins that were detected in all three replicates P5 were included for further analyses, which resulted in a total of 311 proteins identified. Among these proteins, contaminants that are obviously not cytoskeletal proteins were identified from the datasets based on the GO terms and function annotations. For instance, proteins annotated as histone, kinase, or DNA binding proteins, or proteins located in subcellular compartments such as translational apparatus, nucleus, and plasma membrane were removed from the datasets. As a consequence, the number of putative cytoskeletal proteins identified in P5 was reduced to 239. DeepCoil 2.0 was employed to predict coiled-coil domains (ccds) from the 239 putative cytoskeletal proteins based on protein sequence[54]. Three indices, i.e., the number of ccds within each protein, the average length of ccds in each protein, and the percentage of total protein length occupied by ccds, were calculated based on the output of DeepCoil 2.0. In addition to the cytoskeletal proteome in this study, the presence of ccds was also investigated for the hydrogenosome proteome of *T. vaginalis* and a randomly picked *T. vaginalis* protein dataset[55]. A summary of mass spectrometry information is given in Supplementary Table 6, and mass spectrometry P5 protein results are located in the Supplementary Data 1.

### CryoEM sample preparation and image acquisition

To prepare DMTs for single particle analysis, 2.5 μL of DMT lysate was applied to glow discharged carbon holey grids (R2/1) (Ted Pella) and incubated on the grid for 1 min prior to blotting and plunge freezing into a 50:50 mixture of liquid ethane and propane using a Vitrobot Mark IV (Thermo Fisher Scientific). Flash-frozen grids were stored under liquid nitrogen until cryoEM imaging.

Dose fractionated cryoEM movies were recorded on a K3 direct electron detector (Gatan) equipped Titan Krios electron microscope (FEI/Thermo Fisher Scienific) fitted with a Gatan Imaging Filter (GIF) and operated at 300 keV. Movies were recorded at a nominal magnification of 81,000 × and calibrated pixel size of 0.55 Å at the specimen level, operated in super resolution mode. Using SerialEM[56], 30,834 movies were recorded with a cumulative electron dose of ~ 45 e⁻/A².

### CryoEM image processing and 3-dimensional reconstruction

Movie frame alignment and motion correction were performed in CryoSPARC[57] to generate cryoEM micrographs from each movie. Patch-aligned and dose weighted micrographs were binned 2X to improve processing speeds and transferred for processing in Relion 4.0 and Topaz automated particle picking, using the filament option "-f" incorporated by Scheres and colleagues[58–60]. Picked particle coordinates were extracted in Relion using the particle extract job with helical option enabled to extract particles every 8.2 nm along the picked filaments. The extracted particles were transferred back to CryoSPARC for further analysis and 3D reconstruction. 942,986 DMT particles were initially screened for quality using the 2D classification job type, and those classes with good features were chosen for further data processing, leaving 868,683 particles. Initial 3D reconstructions were made using 2X binned particles to expedite data processing. CryoSPARC's Helix refine job type was used to refine the DMT particles

and prevent particles from the same filament from being placed in different half-sets during refinement. With half-sets determined, the particles were then subjected to non-uniform refinement to yield an initial DMT reconstruction based on the 8.2 nm repeating tubulin heterodimer organization.

We next carried out focused classification and refinements, similar to the strategy previously used to further process the *Chlamydomonas* doublet microtubule[19], using CryoSPARC. Briefly, cylindrical masks over MIPS or MOPs with known periodicities were used to relax the 16, 48, and 96 nm periodicity from the initial 8.2 nm repeating DMT structure in a stepwise fashion. To improve local resolutions, we performed focused local refinements wherein cylindrical masks were placed over specific protofilaments so that CryoSPARC could be used to align those protofilaments and their MIP and MOP features. This resulted in 8, 16, 48, and 96 nm reconstructions with 3.8, 3.8, 4.2, and 4.3 Å global resolutions, respectively. Local resolutions were improved using the local refinement job types in CryoSPARC, with maps over the regions of interest. Systematic local refinement was performed along the length of the 48 nm using small masks along the length of the tubules to improve local resolutions to between 3.2 and 4.2 resolution. Locally refined maps were combined in UCSF ChimeraX[62] using the "vol maximum" command on the 35 locally refined maps to generate a composite map.

To resolve the radial spoke (RS) proteins, particles were picked automatically using Topaz trained on RS particles[58]. This picking yielded 131,906 particles, which were extracted and subjected to ab-initio and subsequent non-uniform refinement in CryoSPARC to improve their resolution to 4.7 Å. Local refinements of the head and neck domains increased local resolution to between 4.2 and 5 Å resolution, which enabled fitting of homology models from *C. reinhardtii*[57].

### Atomic modeling and docking

The tubulin models were built using AlphaFold predicted models of α- and β-tubulin and using molecular dynamics flexible fitting software in UCSF ChimeraX[61,62]. To model MIPs, homologs from other organisms with existing structures were roughly fit into our DMT maps before using NCBI's basic local alignment search tool (BLAST) to identify homologs in *Tv* and confirmed their identity using our mass spectrometry data[63]. For densities lacking homologous proteins, initial models were built using DeepTracer[64], followed by refinement in Coot[65].

The identities of unknown densities were confirmed using automated building in ModelAngelo and standard Protein BLAST of the predicted amino acid sequences against the TrichDB database[63,66,67]. Alternatively, or often in combination with ModelAngelo predicted models, cryoID was used to identify the most likely candidates for cryoEM densities[29]. Further attempts to fit proteins in low-resolution regions were made using a strategy similar to that of the DomainFit software package[45]. Briefly, visual inspection of AlphaFold predicted structures also aided in matching of candidates with map density shapes to assess potential matches. Models were fit using Coot and ISOLDE[61,68] and refined using Phenix Real Space Refinement[69].

Docking of thiabendazole (SMILES: C1 = CC = C2C( = C1)NC( = N2) C3 = CSC = N3) into *Tv* β-tubulin was performed using SwissDock tools and AutoDock Vina version 1.2.0[70,71]. Tyrosine was mutated to phenylalanine using the "swap amino acid" function in UCSF ChimeraX[62]. The box center was placed at 361 – 472 – 277 for each run with dimensions 10 – 10 – 15, and sampling exhaustivity set to the default value of 4.

We used the same software as above for docking (SMILES: C1(C(C(C(C(C(C1OP( = O)(O)O)OP( = O)(O)O)OP( = O)(O)O)OP( = O)(O) O)OP( = O)(O)O)O) into *Tv*FAP40, except that the box center was placed at the ligand density and box size was change to 10 – 10 – 10 with sampling exhaustivity of 8. Grid box size was chosen to constrain ligands to putative binding sites from previous studies (thiabendazole) or observed localization (IP5)[32]. To compare the binding affinities of IP5 and trichomonad metabolites, .sdf files of all molecules were

retrieved from PubChem, converted to.pdbqt format in Open Babel 3.1.0, and docked to the *Tv*FAP40 ligand binding pocket using AutoDock Vina[72,73]. Structure visualization and figure preparation were done with UCSF ChimeraX and Adobe illustrator, respectively[62].

### Structural Homolog Search For *Tv*FAP35 and *Tv*FAP40

Structural homologs of *Tv*FAP35 and *Tv*FAP40 in other organisms were identified with FoldSeek using the *Tv*FAP40 kinked-coiled-coil (residues 126–361) as the query. Hits were manually filtered to retain only those that both contained the kinked-coiled-coil motif and aligned to the query with a TM-score ≥ 0.4. Amino acid sequences of structural homologs were retrieved from UniProt and aligned using Clustal Omega[37,74]. Multiple sequence alignments were visualized with ESPript 3.0[75]. Evolutionary relationships between protists were obtained from the NCBI Taxonomy Database and depicted in a phylogenetic tree created in Adobe Illustrator (Supplementary Fig. 5).

### AI-assisted Ligand identification from cryoEM density

A difference map containing only the *Tv*FAP40 ligand was resampled to a grid size of 0.2 Å in UCSF ChimeraX and converted to npz format for compatibility with the LigandRecognizer web server[39], which was used to identify the ligand category.

### Reporting summary

Further information on research design is available in the Nature Portfolio Reporting Summary linked to this article.

## Data availability

The cryoEM map data generated in this study have been deposited in the Electron Microscopy Data Bank (EMDB) under the following accession codes: the 16, 48, and 96 nm repeats under EMD-46642, EMD-46643, and EMD-46636, respectively. The composite cryoEM map of the 48 nm repeat is available under EMD-46580, and the coordinates for the complete atomic models are available under PDB accession code 9D5N. Focused cryoEM maps for the 8 nm *Tv*OJMOP1, 16 nm *Tv*FAP40, and 96 nm N-DRC refinements are available at the EMDB under accession numbers EMD-46634, EMD-46633, and EMD-46635, respectively. Previously published 48 nm repeat doublet structures used as reference models for comparison in our study are 6U42 (*Clamydomonas*), 8OTZ (*Bovine),* and 8G2Z (*Tetrahymena*). Reconstruction of the *Tv*RS head and neck is available under EMDB accession number EMD-48446. The mass spectrometry data used in this study have been deposited in the MassIVE database under accession code: MSV000096489 [10.25345/C5DJ58V1N].

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

## Acknowledgements
We thank Yanxiang Cui for his preliminary effort on this project, Xian Xia, and Kent Hill for their expert advice in the development of this work. This work was supported in part by grants from the National Institutes of Health (R01GM071940 to Z.H.Z. and R01AI103182/R33AI119721 to P.J.J.). K.A.M. and A.S. received support from the NIH Ruth L. Kirschtein National Research Service Award AI007323. We acknowledge the use of resources at the Electron Imaging Center for Nanomachines, supported by UCLA and by instrumentation grants from NIH (1S10RR23057, 1S10OD018111) and NSF (DBI-1338135 and DMR-1548924). We acknowledge support from the UCLA AIDS Institute, the James B. Pendleton Charitable Trust, and the McCarthy Family Foundation.

## Author contributions
Z.H.Z. and P.J. designed and supervised the project. K.A.M., S.E.W., and A.S. prepared samples. S.E.W. conducted mass spectrometry work. A.S. and S.K. performed cryoEM imaging and prepared 3D reconstructions. Under the guidance of Z.H.Z., A.S., S.K., and E.H.C. built the atomic models, interpreted the structures, made the figures, and wrote the paper; all authors reviewed and approved the paper.

## Competing interests
The authors declare no competing interests.
