## [Transparent Peer Review file · Nature Communications]

Structures of Native Doublet Microtubules from *Trichomonas vaginalis* Reveal Parasite-Specific Proteins

Corresponding Author: Professor Z. Hong Zhou

Version 0:

Reviewer comments:

Reviewer #1

(Remarks to the Author)

This work by A. Stevens et al. reported a new microtubule doublet structure from *Trichomonas vaginalis*. The authors employed single-particle cryo-EM, mass spectrometry, and computational approaches to determine and build the atomic models of Tv-DMT. This allowed them to compare the structures with DMTs from other species, providing important insights into the structural basis of Tv motility. These findings could potentially contribute to the development of new drugs targeting the Tv flagella. The structure itself is beautiful and certainly worthy of publication. I have no major concerns about its acceptance in general. However, I do have a few comments on clarity and concerns regarding some overstatements in the manuscript. While it is good to emphasize its potential implications in drug design and structure-based medical treatments, this might not be the primary merit of the work. I would suggest the authors tone down this point and instead focus more on their potential impact on flagellar assembly and the mechanism of flagellar beating.

Minor comments:

1. In the abstract, the authors stated that they identified 30 unique proteins, including 19 microtubule inner proteins and 9 microtubule outer proteins. This is somewhat confusing, as 19 plus 9 does not equal 30. I assume the authors included tubulins in this count, but this doesn't seem appropriate for stating they 'identified' 30 proteins. In fact, only 4 of these proteins are novel. The authors should be more careful with their use of the term 'identification' in the abstract and other sections throughout the manuscript.
2. The microtubule doublet structure is certainly a central target in the flagellar system. Although several reports exist on doublet structures from various species, a deeper mechanistic understanding of this system remains limited, primarily due to the lack of high-resolution structures of dynein, the key molecule driving flagellar movement and swimming through the human genitourinary tract. Any insights into the structure, assembly, and functional states of outer-arm dynein during the flagellar beating cycle would be invaluable. Unfortunately, this critical question is not addressed or even discussed in this manuscript.
3. 'We optimized a protocol to isolate DMTs from *T. vaginalis*.' The authors need to cite the original protocol here or briefly explain the specific type of protocol used in more detail.
4. IP6: While the density fits with IP6, I wouldn't say it is a perfect or the only possible match. It would be better if the authors could at least propose alternative possibilities or compare IP6 with other similar ligands. A clear comparison between the top and second hits would add clarity. Additionally, targeting this pocket may not be a promising approach, as IP6 is known to bind several human proteins.
5. '... human-borne parasite, it has demonstrated the power of in situ cryo-EM over other structural ...'. This statement is somewhat confusing. The authors solved the doublet structure using isolated samples, not in the native environment, but rather from a native source. Therefore, it would be inappropriate to make this overstatement, and this point should be revised accordingly.

Reviewer #2

(Remarks to the Author)

The manuscript from Alexander Stevens et al. presents a comprehensive analysis of *Trichomonas vaginalis* (Tv) flagellar doublet microtubules (DMTs), identifying both conserved and novel microtubule inner proteins (MIPs) and outer proteins (MOPs). The study makes excellent use of cryo-electron microscopy (cryo-EM) to generate high-resolution structural data and compares these structures with previously characterized organisms. The identification of novel MIPs and MOPs provides valuable insights into the structural differences that may underlie species-specific variations in microtubule assembly and flagellar function. Additionally, the identification of Tv-specific proteins that might play roles in drug resistance and DMT stabilization significantly contributes to the fields of parasitology and microtubule biology. Overall, this is a highly innovative study that lays a solid foundation for further exploration of microtubule-associated proteins in parasitic organisms.

Structural insights: the comparison between Tv-tubulin and human tubulin, particularly in the context of drug resistance, is insightful. The detailed analysis of the interaction between benzimidazole drugs and Tv-tubulin reveals important structural differences that could influence drug efficacy and parasite resistance.

Discovery of Novel Proteins: the identification of novel MIPs and MOPs, specifically TvFAP40 and TvFAP35, and the detailed characterization of their structural features and potential roles in DMT stability, flagellar beating, and drug interactions, is significant. The demonstration that TvFAP40 could be a druggable target is especially important considering the global health impact of *Trichomonas vaginalis*.

Major comments

While the work presents the first high-resolution view of Tv DMTs, it would be valuable to include more details on the protein composition of the samples used (P4, P5 samples). This would support the conclusion that the A-tubule is simplistic compared to DMTs in other organisms. Is this simplicity expected in other protists? Additional clarification here would strengthen the argument.

The computational docking studies suggest that specific residues of TvFAP40 may interact with an inositol polyphosphate (presumably IP6), potentially making TvFAP40 an intriguing drug target. However, the lack of experimental validation (e.g., mutagenesis experiments to confirm IP6 binding) weakens this conclusion. Incorporating such experiments would further solidify the study's conclusions. To a lesser extent, this (experimental validation) also applies to the role of TvFAP40 and TvFAP35 in DMT assembly and stabilization suggested by the authors.

Minor comments.

Abstract

Line 10. Since no functional data are presented for the role of TvFAP40 and 35 in DMT stability and flagellar beating, it would be more accurate to tone down the sentence.

Introduction

Line 29-32: It would be helpful to mention if there is experimental evidence in the literature that supports the role of DMT stabilization in *T. vaginalis*

Line 35-37: A recent study by Zhou et al. (2023) detailing MIPs and MOPs in the sperm flagellum (Zhou, L. et al. Cell 186, 2897-2910.e19) should be cited to provide a broader context.

Line 50: Consider replacing "IA" with modeling

Line 56-58: The notion of "same environment but different number of flagella" should be integrated into the sentence for improved readability.

Results:

Line 61. Please specify the optimization steps used in your protocol compared to previous protocols and how these optimizations reduce perturbations in DMT structures

Line 72. The work of Andersen et al. (Nat Commun 15, 2687 (2024), which identifies shared motifs in non-homologous proteins conserved in flagellar structures, may offer insights into your proteomic data. Does your P5 sample contain protein sequences that were not assigned but that align with these motifs?

Line 85: Could the authors provide a reference for BZ assays in *T. vaginalis* to support species-specific sensitivity to benzimidazoles?

Line 105. Clarify if "globular MIPs that span PFs B3-B4 and B5-B6 and exhibit 96 nm periodicity" correspond to B-MIP1 and B-MIP2 in Figure 2b. If so, consistent nomenclature between the text and figures is recommended.

Line 112. Are all four Tv flagella equivalent in terms of MIPs and MOPs? Please provide commentary.

Line 130-137. Provide an explanation of how these interactions reinforce the inner junction for non-specialist readers.

Line 181. Specify that the structure being referred to is from *Tetrahymena*.

Line 189. Given the paragraph's focus, consider citing Andersen et al. (2024).

Line 197. Proteomic data are not provided. Including a supplementary table with these data, or at least the 31 shortlisted proteins, would add transparency. Also, what are the differences in protein content between P4 and P5 samples? A protein list for each sample would clarify this.

Line 200. Please provide the accession numbers for the six kinked-coiled-coil homologs from protists.

Line 228 and figure 6. There is confusion in the nomenclature of proteins throughout the manuscript and figures. Please ensure consistent naming (e.g., TvMPO1, TvMPO2, TvOJMOP1, etc.). Also, figure 6 legend (f and g) seems inverted—please clarify.

Discussion:

Line 231. The hypothesis that TvOJMOP1 may be lost during DMT preparation—do the proteomic analyses of P4 and P5 support this?

Line 252. Clarify that most of the MIPs and MOPs described are conserved across ciliated organisms to strengthen the claim that these proteins mediate DMT function in flagella.

Line 264-269. The fact that Tv has five flagella suggests potential differences in flagellar beating compared to the sperm flagellum—commenting on this would be valuable.

Line 288. Specify whether the homologs are sequence or structural homologs and whether they are flagellar proteins in other organisms. Additionally, it would be interesting to discuss whether tektin is present in these organisms. Are both TvFAP40 and TvFAP35 present, or only one?

Material and methods

Line 419. Typo—refinement spelling correction

Figure legends.

Figure 1b. A color code similar to that used in Figure 2b would aid the reader

Figure 6h. Include a color code.

Reviewer #3

(Remarks to the Author)

General assessment

In this manuscript, Stevens et al. revealed the ultrastructure of *Trichomonas vaginalis* (Tv) doublet microtubule (DMT) for the first time by high-resolution cryo-EM analysis. Tv causes human trichomoniasis, the most common non-viral sexually transmitted disease. This study provides a structural basis for the rational design of trichomoniasis therapeutics. Remarkably, the authors identified four novel MIPs proteins in Tv DMT, which can be a potential drug target, reinforcing this study's significance.

The authors' detailed description of the Tv DMT structures also enhances this report's value. Comparison of DMT structures from other species revealed the diversity of DMT MIPs arrangement and suggested minimum MIPs combination for flagella function. Tv β -tubulin structure docked with thiabendazole clearly shows the mechanisms that the reagent works as an antiparasitic. Moreover, they identified a novel MT-binding motif based on the Tv DMT proteins they found: TvFAP40 and TvFAP35.

In conclusion, this study offers significant insights into the comprehensive structure of Tv DMT, which can serve as a structural framework for understanding the rational drug design / the drug resistance of trichomoniasis. However, certain aspects of the results and discussion require further clarification and improvement to comprehend the authors' statements, especially for IP6 identification as a TvFAP40 ligand.

Main points

(1) General information for Tv flagella structure is not provided.

(1)-1

For Tv locomotion, not only DMT but also its appendages (ODA, IDA, and Radial Spoke) have essential roles in driving or modifying the flagella movement. However, this manuscript does not show any structure of Tv ODA, IDA, and RS. Recent studies have revealed the diversity of these appendages, such as the difference in OAD motor-domain numbers, OAD docking-complex structures, and RS arrangement. The reviewer recognizes that this manuscript is focused on the DMT MIPs and MOPs. Still, providing the rough image of 96nm Tv DMT with ODA, IDA, and RS helps readers understand the

whole Tv axoneme architecture.

(1)-2

The raw EM image of Tv DMT is neither provided in this manuscript nor deposited in EMPIAR, making it hard to assess the sample quality and purity. In addition, it is not clarified whether the Tv's two types of flagella (four anterior flagella and one membrane-bound flagellum) are purified separately.

(1)-3

Raw results of Mass Spectrometry of Tv DMT are not provided in this manuscript, nor referred to in the "Data availability" section.

(2) The identification process of IP6 as a TvFAP40 ligand needs more clarified explanations.

(2)-1

Although the TvFAP40 ligand density is described as a "six-pointed star-shaped ligand," there is no figure image that clearly shows the ligand structure from the front side. Ligand maps superimposed on Fig. 4e, g, and h will help readers validate the fitting of IP6 as a TvFAP40 ligand.

(2)-2

The paragraph in lines 150-163 is hard to follow, since this paragraph is a mixture of "Results" and "Discussion". The main results in this paragraph are the *in silico* molecular docking of IP6 and TvFAP40. However, this paragraph also contains the known IP6 functions in HIV and zebrafish, which should be mentioned in the "Discussion" section. Similarly, the speculated TvFAP40-IP6 functions mentioned in lines 157-159 should also be discussed in the "Discussion" section.

>Lines 157-159: "These results support the notion that IP6 acts as a ligand within TvFAP40 and may stabilize the head to reinforce its interactions with enkurin and the tail of its neighboring monomer and microtubule PF."

(2)-3

Related to (2)-2, the section title in line 138 is misleading: "Tv-specific FAP40 head domain binds a stabilizing ligand". The stabilizing function of the TvFAP40 ligand is not validated in this manuscript, but just based on the authors' speculation. To avoid overstatement, the authors should modify the section title based on their results.

Reviewer #4

(Remarks to the Author)

Version 1:

Reviewer comments:

Reviewer #1

(Remarks to the Author)

The authors addressed my questions clearly. The manuscript has been substantially improved. I have no more concerns.

Reviewer #2

(Remarks to the Author)

The authors of the manuscript entitled "Structures of Native Doublet Microtubules from *Trichomonas vaginalis* Reveal Parasite-Specific Proteins" have addressed the majority of our comments and revision requests. The clarity of the manuscript has been significantly improved, and we believe it is now suitable for publication.

Reviewer #3

(Remarks to the Author)

The revised manuscript enhances the description and interpretation of Tv's DMT structure. The authors included the resolved structures of Tv's N-DRC and RS. Although these maps remain at a medium resolution level, they reveal several species-specific features that can contribute to understanding the axonemal diversity across various waveforms. Moreover, the authors refined their interpretations of TvFAP40's ligand. Providing several rationales, the authors suggest that IP5 corresponds to the ligand density. They also mention other ligand candidates in Figure S3 and related sections. However, Figure S3 contains several misprints: (1) In Figures S3e and S3f, the label states IP6 instead of IP5; (2) Figure S3g, annotated on lines 176 and 178, does not display. The authors need to address these points. Nonetheless, the rest of the manuscript features high-quality cryo-EM data, and the interpretations are reasonable and satisfactory. This is commendable and significant work.

Reviewer #4

(Remarks to the Author)

Responses to Reviewers' Comments

Summary of responses

We thank the three reviewers for the prompt consideration and unanimous support of the paper. While most suggestions are minor, there are two main suggestions from the reviewers: The **first** concerns the DMT bound dynein regulatory components (DRC) and radial spoke (RS) structures (Reviewers 1 and 3); The **second** concerns the interpretation of FAP40-bound density as IP6 (all three reviewers). We have taken the last several months to improve our structure and modeling in an effort to address both issues. For the **first** issue, we have improved the map to resolve DRC and RS and have expanded our text reporting these structures; medium resolution structures of DRC and RS are now included in the revised Supplementary Figure S1 and the new Supplementary Figure S5 and S1. To address the **second** concern, we have significantly toned down the interpretation of the TvFAP40-bound ligand as a sugar-based molecule, with IP5 being a promising candidate. The focus has shifted to the structural features of the binding pocket, with added docking analyses and ligand classification results presented in new tables and expanded methods. This provides a more balanced discussion of the TvFAP40 ligand's role.

For your convenience of perusing this document, we have copied the reviewer's critique in **Black** and our responses are shown in **blue** color. You will see that all comments are addressed carefully and corresponding suggestions incorporated into the revised manuscript without tracked changes and highlighted in combined document.

Reviewer #1

Summary by Reviewer #1: This work by A. Stevens et al. reported a new microtubule doublet structure from *Trichomonas vaginalis*. The authors employed single-particle cryo-EM, mass spectrometry, and computational approaches to determine and build the atomic models of Tv-DMT. This allowed them to compare the structures with DMTs from other species, providing important insights into the structural basis of Tv motility. These findings could potentially contribute to the development of new drugs targeting the Tv flagella. The structure itself is beautiful and certainly worthy of publication. I have no major concerns about its acceptance in general. However, I do have a few comments on clarity and concerns regarding some overstatements in the manuscript. While it is good to emphasize its potential implications in drug design and structure-based medical treatments, this might not be the primary merit of the work. I would suggest the authors tone down this point and instead focus more on their potential impact on flagellar assembly and the mechanism of flagellar beating.

Response: We thank the reviewer for recognizing the merits of this work and pointing out the weakness of those statements related to drug development. We have toned down the potential implications of this work in drug design which is reflected throughout the paper and in a change to the title.

Minor comments:

1. In the abstract, the authors stated that they identified 30 unique proteins, including 19 microtubule inner proteins and 9 microtubule outer proteins. This is somewhat confusing, as 19 plus 9 does not equal 30. I assume the authors included tubulins in this count, but this doesn't seem appropriate for stating they 'identified' 30 proteins. In fact, only 4 of these proteins are novel. The authors should be more careful with their use of the term 'identification' in the abstract and other sections throughout the manuscript.

Response: We have altered the abstract and subsequent portions of the manuscript to remove this ambiguity through more direct language, e.g. using 'different' in place of 'unique', and 'observed' in place of 'identified' when referring to the complete organization to avoid misleading the reader (Abstract lines 29-30).

2. The microtubule doublet structure is certainly a central target in the flagellar system. Although several reports exist on doublet structures from various species, a deeper mechanistic understanding of this system remains limited, primarily due to the lack of high-resolution structures of dynein, the key molecule driving flagellar movement and swimming through the human genitourinary tract. Any insights into the structure,

assembly, and functional states of outer-arm dynein during the flagellar beating cycle would be invaluable. Unfortunately, this critical question is not addressed or even discussed in this manuscript.

Response. We did not resolve structures of the dynein motor proteins to reveal species-specific variations. However, we have added sections to the results related to this topic and accompanying supplementary figure to discuss the NDRC complex (lines 243-253 and Fig.S5). Further, we resolved a medium-resolution reconstruction of the radial spoke proteins (Fig. S5), which connect the DMTs to the central pair, and show species-specific variations which may inform our understanding of the axonemal superstructure.

3. 'We optimized a protocol to isolate DMTs from *T. vaginalis*.' The authors need to cite the original protocol here or briefly explain the specific type of protocol used in more detail.

Response. We have now cited the original protocol. See Line 84.

4. IP6: While the density fits with IP6, I wouldn't say it is a perfect or the only possible match. It would be better if the authors could at least propose alternative possibilities or compare IP6 with other similar ligands. A clear comparison between the top and second hits would add clarity. Additionally, targeting this pocket may not be a promising approach, as IP6 is known to bind several human proteins.

Response: We have toned down our assertion of IP6 as the best fit and now more broadly suggest a potential sugar ligand, with our example being IP5. We now provide a table of docked molecules from Autodock Vina (Table S5), and the Ligand classification tool (Table S4) we used to guide our assertion. This process is now described in detail in the methods section (lines 497-500). We believe the unique nature of the binding pocket, not the ligand, make *TvFAP40* an interesting target. We have altered language throughout the text to reflect this.

5-1. '... human-borne parasite, it has demonstrated the power of in situ cryo-EM over other structural ...'. This statement is somewhat confusing. The authors solved the doublet structure using isolated samples, not in the native environment, but rather from a native source. Therefore, it would be inappropriate to make this overstatement, and this point should be revised accordingly.

Response Line 334-336. We removed the term 'in situ' as suggested.

Reviewer #2

Summary by Reviewer #2: The manuscript from Alexander Stevens et al. presents a comprehensive analysis of *Trichomonas vaginalis* (*Tv*) flagellar doublet microtubules (DMTs), identifying both conserved and novel microtubule inner proteins (MIPs) and outer proteins (MOPs). The study makes excellent use of cryo-electron microscopy (cryo-EM) to generate high-resolution structural data and compares these structures with previously characterized organisms. The identification of novel MIPs and MOPs provides valuable insights into the structural differences that may underlie species-specific variations in microtubule assembly and flagellar function. Additionally, the identification of *Tv*-specific proteins that might play roles in drug resistance and DMT stabilization significantly contributes to the fields of parasitology and microtubule biology. Overall, this is a highly innovative study that lays a solid foundation for further exploration of microtubule-associated proteins in parasitic organisms.

Structural insights: the comparison between *Tv*-tubulin and human tubulin, particularly in the context of drug resistance, is insightful. The detailed analysis of the interaction between benzimidazole drugs and *Tv*-tubulin reveals important structural differences that could influence drug efficacy and parasite resistance.

Discovery of Novel Proteins: the identification of novel MIPs and MOPs, specifically *TvFAP40* and *TvFAP35*, and the detailed characterization of their structural features and potential roles in DMT stability, flagellar beating, and drug interactions, is significant. The demonstration that *TvFAP40* could be a druggable target is especially important considering the global health impact of *Trichomonas vaginalis*.

Major comments

1. While the work presents the first high-resolution view of Tv DMTs, it would be valuable to include more details on the protein composition of the samples used (P4, P5 samples). This would support the conclusion that the A-tubule is simplistic compared to DMTs in other organisms. Is this simplicity expected in other protists? Additional clarification here would strengthen the argument.

Response: Our DMT isolation procedure is gentle in order to preserve weakly-associated microtubule associated proteins which leads to heterogeneous samples. Indeed, the P5 fraction in which these DMTs are found had 311 proteins identified, a number far greater than those resolved in this cryoEM map. This ensures enhanced preservation of microtubule associated proteins as evidenced by the presence of the radial spokes for which we resolved a middle-resolution reconstruction to address reviewer 1's concerns (Fig S5). Considering an external species like the radial spokes are well preserved on the surface of DMT support the notion that our isolation was gentle enough to preserve most MIP species. It is unclear if other protists will be as simple as *T. vaginalis* but others such as *T. brucei* similarly lack tektin proteins (Berriman, M. et al. 2005). We acknowledge this in the updated results section lines 224-225.

2-2. The computational docking studies suggest that specific residues of TvFAP40 may interact with an inositol polyphosphate (presumably IP6), potentially making TvFAP40 an intriguing drug target. However, the lack of experimental validation (e.g., mutagenesis experiments to confirm IP6 binding) weakens this conclusion. Incorporating such experiments would further solidify the study's conclusions. To a lesser extent, this (experimental validation) also applies to the role of TvFAP40 and TvFAP35 in DMT assembly and stabilization suggested by the authors.

Response: We have toned down our assertions that an inositol based ligand candidate and now use IP5 as an example candidate along with other ligands (Fig. S3). We lack the resolution to conclusively identify this ligand, however we now detail the method used to identify ligand candidates in a less biased manner using newly available ligand classification software for cryoEM maps and docking.

As *Tv* is not a model organism, genetic manipulations are not straightforward. Indeed, our co-author Dr. Patricia Johnson is a world-leading expert in the field of *Tv* molecular biology, but our efforts to knock out or mutate these proteins proved unsuccessful. As such we've toned down our language with regards to the putative roles of TvFAP40 and TvFAP35 in DMT stabilization throughout the piece.

Minor comments.

Abstract

1. Line 10. Since no functional data are presented for the role of TvFAP40 and 35 in DMT stability and flagellar beating, it would be more accurate to tone down the sentence.

Response: Line 33-34. "to stabilize DMTs and enable *Tv* locomotion" changed to "and interface with stabilizing MIPs".

Introduction

2. Line 29-32: It would be helpful to mention if there is experimental evidence in the literature that supports the role of DMT stabilization in *T. vaginalis*

Response: Line 60-64. Cited additional studies to clarify the role of conserved MIPs in DMT stabilization.

3. Line 35-37: A recent study by Zhou et al. (2023) detailing MIPs and MOPs in the sperm flagellum (Zhou, L. et al. Cell 186, 2897-2910.e19) should be cited to provide a broader context.

Response: Line 59. Citation introduced as suggested.

4. Line 50: Consider replacing "IA" with modeling

Response: Line 72. We prefer the term “artificial intelligence” as we leveraged many AI enabled applications, such as AlphaFold, FoldSeek, Autodock Vina among others to build and analyze the DMT structures.

5. Line 56-58: The notion of "same environment but different number of flagella" should be integrated into the sentence for improved readability.

Response: Lines 78-80. Updated to reflect the difference in flagella number between the organisms.

Results:

6. Line 61. Please specify the optimization steps used in your protocol compared to previous protocols and how these optimizations reduce perturbations in DMT structures

Response: Lines 83-84. Protocol from previously published results has been cited as suggested by reviewer 1.

7. Line 72. The work of Andersen et al. (Nat Commun 15, 2687 (2024)), which identifies shared motifs in non-homologous proteins conserved in flagellar structures, may offer insights into your proteomic data. Does your P5 sample contain protein sequences that were not assigned but that align with these motifs?

Response: Our proteomic data contains protein sequences that could be fit into the unassigned densities, but the candidate sequences often had orthologs with similar sequences and predicted structures. Since we lacked sufficient resolution to compare fitting of side chains for these densities, we did not feel confident assigning proteins to them.

8. Line 85: Could the authors provide a reference for BZ assays in *T. vaginalis* to support species-specific sensitivity to benzimidazoles?

Response: Line 100-101. "*In vitro*" effect of microtubule inhibitors on *Trichomonas vaginalis* by Juliano et. al. 1985 documents the sensitivity of *Tv* to thiabendazole among other microtubule inhibitors. This work is referenced in the present article.

9. Line 105. Clarify if "globular MIPs that span PFs B3-B4 and B5-B6 and exhibit 96 nm periodicity" correspond to B-MIP1 and B-MIP2 in Figure 2b. If so, consistent nomenclature between the text and figures is recommended.

Response: Line 128. Text was updated to clarify that the density in question was B-MIP1.

10. Line 112. Are all four *Tv* flagella equivalent in terms of MIPs and MOPs? Please provide commentary.

Response: Our purification method yielded a sample where the different *Tv* flagella could not be distinguished. Nothing during data processing suggested significantly different global architectures. A suggestion to this end was added to explain the variable presence of OJMOP1 Line 266-270.

11. Line 130-137. Provide an explanation of how these interactions reinforce the inner junction for non-specialist readers.

Response: Lined 156-158. Brief description of known role of Enkurin in DMT assembly and stability to suggest why securing this protein may stabilize inner junction in *Tv*.

12. Line 181. Specify that the structure being referred to is from *Tetrahymena*.

Response: Lines 204-206. Specified that FAP77 was found to aid in B-tubule formation specifically in *Tetrahymena*, and delineated description of FAP77 in *Tv*.

13. Line 189. Given the paragraph's focus, consider citing Andersen et al. (2024).

Response: Lines 213-214 incorporated to cite acknowledge work by Andersen and improve context.

14. Line 197. Proteomic data are not provided. Including a supplementary table with these data, or at least the 31 shortlisted proteins, would add transparency. Also, what are the differences in protein content between P4 and P5 samples? A protein list for each sample would clarify this.

Response: Line 384-385. We have added a table with the UniProt IDs of all 31 proteins as well as an XML file of our complete P5 results. As only P5 was used for this paper, we removed references to P4 in our methods section to reduce confusion. The data related to P4 is being used for another ongoing experiment. We have uploaded the MS analysis of P5 to the MassIVE database which is now available for your review. MassIVE database. Username: MSV000096747_reviewer Password: 7Zo20S9boDgXG82x

15. Line 200. Please provide the accession numbers for the six kinked-coiled-coil homologs from protists.

Response: Line 226 Added as suggested.

16. Line 228 and figure 6. There is confusion in the nomenclature of proteins throughout the manuscript and figures. Please ensure consistent naming (e.g., TvMPO1, TvMPO2, TvOJMOP1, etc.). Also, figure 6 legend (f and g) seems inverted—please clarify. other

Response: We removed the *Tv* prefix from all proteins but FAP35 and FAP40 to improve clarity. Fig. 6f-g legend corrected as suggested to enhance clarity.

Discussion:

17. Line 231. The hypothesis that TvOJMOP1 may be lost during DMT preparation—do the proteomic analyses of P4 and P5 support this?

Response: We have provided an additional explanation, that OJMOP1 may not be present on all flagella and explain the different classes of particles (lines 266-270). We are no longer using P4 to analyze this data, but it is unclear whether we could derive any conclusions from these fairly heterogeneous samples.

18. Line 252. Clarify that most of the MIPs and MOPs described are conserved across ciliated organisms to strengthen the claim that these proteins mediate DMT function in flagella.

Response: Lines 290-292. Change incorporated as suggested.

19. Line 264-269. The fact that *Tv* has five flagella suggests potential differences in flagellar beating compared to the sperm flagellum—commenting on this would be valuable.

Response: Lines 306-309. Concept of multi-flagellated vs mono-flagellated propulsion mentioned as suggested.

20. Line 288. Specify whether the homologs are sequence or structural homologs and whether they are flagellar proteins in other organisms. Additionally, it would be interesting to discuss whether tektin is present in these organisms. Are both TvFAP40 and TvFAP35 present, or only one?

Response: updated line 331 to mention “structurally homologous” instead of just “homologous.” Also added TvFAP35 into the description of structural homologs on line 324.

Material and methods

21. Line 419. Typo—refinement spelling correction

Response: Line 463: corrected the spelling of “refinement” on line 463.

Figure legends.

22. Figure 1b. A color code similar to that used in Figure 2b would aid the reader

Response: Figure 1 provides an overview of the decorated DMT superstructure but is not intended to portray its fine details. Figure 2 should be sufficient to communicate the detailed MIP arrangement of the *Tv*-DMT.

23. Figure 6h. Include a color code.

Response: Color key added as suggested.

Reviewer #3

Summary by Reviewer #3: In this manuscript, Stevens et al. revealed the ultrastructure of *Trichomonas vaginalis* (Tv) doublet microtubule (DMT) for the first time by high-resolution cryo-EM analysis. Tv causes human trichomoniasis, the most common non-viral sexually transmitted disease. This study provides a structural basis for the rational design of trichomoniasis therapeutics. Remarkably, the authors identified four novel MIPs proteins in Tv DMT, which can be a potential drug target, reinforcing this study's significance.

The authors' detailed description of the Tv DMT structures also enhances this report's value. Comparison of DMT structures from other species revealed the diversity of DMT MIPs arrangement and suggested minimum MIPs combination for flagella function. Tv β -tubulin structure docked with thiabendazole clearly shows the mechanisms that the reagent works as an antiparasitic. Moreover, they identified a novel MT-binding motif based on the Tv DMT proteins they found: TvFAP40 and TvFAP35.

In conclusion, this study offers significant insights into the comprehensive structure of Tv DMT, which can serve as a structural framework for understanding the rational drug design / the drug resistance of trichomoniasis. However, certain aspects of the results and discussion require further clarification and improvement to comprehend the authors' statements, especially for IP6 identification as a TvFAP40 ligand.

Major comments

Main points

1. General information for Tv flagella structure is not provided.

For Tv locomotion, not only DMT but also its appendages (ODA, IDA, and Radial Spoke) have essential roles in driving or modifying the flagella movement. However, this manuscript does not show any structure of Tv ODA, IDA, and RS. Recent studies have revealed the diversity of these appendages, such as the difference in OAD motor-domain numbers, OAD docking-complex structures, and RS arrangement. The reviewer recognizes that this manuscript is focused on the DMT MIPs and MOPs. Still, providing the rough image of 96nm Tv DMT with ODA, IDA, and RS helps readers understand the whole Tv axoneme architecture.

Revisions: We were unable to confidently resolve the structure of the ODA or IDA, however we resolved a medium-resolution structure of the radial spoke and incorporated it into a new diagram of axoneme architecture to improve readers comprehension of the global assembly (Figs. S1 and S5). Further, we now more thoroughly acknowledge the role of other complexes within the axoneme throughout the paper.

2. The raw EM image of Tv DMT is neither provided in this manuscript nor deposited in EMPIAR, making it hard to assess the sample quality and purity. In addition, it is not clarified whether the Tv's two types of flagella (four anterior flagella and one membrane-bound flagellum) are purified separately.

Response: We have added a supplementary figure including an example electron micrograph with microtubule doublets and radial spokes clearly defined (Fig. S1). Our preparation uses a cell disruption in a mild detergent to strip the cellular membranes so it is likely these DMTs come from all flagella. We now acknowledge this throughout the manuscript (e.g. Line 266-270)

3. Raw results of Mass Spectrometry of Tv DMT are not provided in this manuscript, nor referred to in the "Data availability" section.

Response: This data is now available on the MassIVE database under accession number MSV000096747. Username: MSV000096747_reviewer Password: 7Zo20S9boDgXG82x. Please see new supplementary tables and material.

4. The identification process of IP6 as a TvFAP40 ligand needs more clarified explanations.

Response: We no longer assert this ligand is IP6, but instead allude to it likely being a ligand with sugar moiety with shape similar to IP5. Our rationale is explained in the results and the methods (lines 175-188 and lines 506-507).

5. Although the TvFAP40 ligand density is described as a "six-pointed star-shaped ligand," there is no figure image that clearly shows the ligand structure from the front side. Ligand maps superimposed on Fig. 4e, g, and h will help readers validate the fitting of IP6 as a TvFAP40 ligand.

Response: Fig. 4h was modified to emphasize the shape of the TvFAP40-bound ligand

6. The paragraph in lines 150-163 is hard to follow, since this paragraph is a mixture of "Results" and "Discussion". The main results in this paragraph are the in silico molecular docking of IP6 and TvFAP40. However, this paragraph also contains the known IP6 functions in HIV and zebrafish, which should be mentioned in the "Discussion" section. Similarly, the speculated TvFAP40-IP6 functions mentioned in lines 157-159 should also be discussed in the "Discussion" section.

>Lines 157-159: "These results support the notion that IP6 acts as a ligand within TvFAP40 and may stabilize the head to reinforce its interactions with enkurin and the tail of its neighboring monomer and microtubule PF."

Response: Our interpretations have been modified to reflect our uncertainty regarding the ligand identity and these lines have been moved to the discussion section. (lines 316-323)

7. Related to (2)-2, the section title in line 138 is misleading: "Tv-specific FAP40 head domain binds a stabilizing ligand". The stabilizing function of the TvFAP40 ligand is not validated in this manuscript, but just based on the authors' speculation. To avoid overstatement, the authors should modify the section title based on their results.

Response: We have updated the section title to "Tv-specific FAP40 head domain binds an unidentified ligand" (line 163).

References:

1. Berriman, M. et al. The Genome of the African Trypanosome *Trypanosoma brucei*. **Science** 309, 416–422 (2005).

Responses to Reviewers' Comments

Summary of responses

We thank the editor and reviewers for consideration of our paper and prompt feedback of the initial revisions we made. We have copied the reviewer's feedback in **Black** and our responses are shown in **blue** color. All comments are addressed carefully and corresponding suggestions incorporated into the revised manuscript without tracked changes and highlighted in combined document.

Reviewer #1 (Remarks to the Author):

The authors addressed my questions clearly. The manuscript has been substantially improved. I have no more concerns.

We thank the reviewer for their feedback.

Reviewer #2 (Remarks to the Author):

The authors of the manuscript entitled "Structures of Native Doublet Microtubules from *Trichomonas vaginalis* Reveal Parasite-Specific Proteins" have addressed the majority of our comments and revision requests. The clarity of the manuscript has been significantly improved, and we believe it is now suitable for publication.

We thank the reviewer for their feedback.

Reviewer #3 (Remarks to the Author):

The revised manuscript enhances the description and interpretation of Tv's DMT structure. The authors included the resolved structures of Tv's N-DRC and RS. Although these maps remain at a medium resolution level, they reveal several species-specific features that can contribute to understanding the axonemal diversity across various waveforms. Moreover, the authors refined their interpretations of TvFAP40's ligand. Providing several rationales, the authors suggest that IP5 corresponds to the ligand density. They also mention other ligand candidates in Figure S3 and related sections. However, Figure S3 contains several misprints: (1) In Figures S3e and S3f, the label states IP6 instead of IP5; (2) Figure S3g, annotated on lines 176 and 178, does not display. The authors need to address these points. Nonetheless, the rest of the manuscript features high-quality cryo-EM data, and the interpretations are reasonable and satisfactory. This is commendable and significant work.

We have updated the label to IP5 instead of IP6 in Figures S3e and S3f, now referenced as Supplemental Figures 4e and 4f in the manuscript. As Figure S3g was not displayed before, we now have included it as Supplementary Figure 4g and this is properly referenced in lines 160-162 of the manuscript.

Reviewer #4 (Remarks to the Author):

We thank the reviewer for their feedback.